# Nicotinamide riboside is uniquely and orally bioavailable in mice and humans

Samuel A.J. Trammell[1], Mark S. Schmidt[1], Benjamin J. Weidemann[1], Philip Redpath[2], Frank Jaksch[3], Ryan W. Dellinger[3], Zhonggang Li[4], E Dale Abel[1,4], Marie E. Migaud[1,2] & Charles Brenner[1,4]

Nicotinamide riboside (NR) is in wide use as an $NAD^+$ precursor vitamin. Here we determine the time and dose-dependent effects of NR on blood $NAD^+$ metabolism in humans. We report that human blood $NAD^+$ can rise as much as 2.7-fold with a single oral dose of NR in a pilot study of one individual, and that oral NR elevates mouse hepatic $NAD^+$ with distinct and superior pharmacokinetics to those of nicotinic acid and nicotinamide. We further show that single doses of 100, 300 and 1,000 mg of NR produce dose-dependent increases in the blood $NAD^+$ metabolome in the first clinical trial of NR pharmacokinetics in humans. We also report that nicotinic acid adenine dinucleotide (NAAD), which was not thought to be en route for the conversion of NR to $NAD^+$, is formed from NR and discover that the rise in NAAD is a highly sensitive biomarker of effective $NAD^+$ repletion.

[1] Department of Biochemistry, Carver College of Medicine, University of Iowa, Iowa City, Iowa 52242, USA. [2] John King Laboratory, School of Pharmacy, Queens University Belfast, Belfast BT7 1NN, UK. [3] ChromaDex, Inc., 10005 Muirlands Blvd, Suite G, Irvine, California 92618, USA. [4] Department of Internal Medicine, Carver College of Medicine, University of Iowa, Iowa City, Iowa 52242, USA. Correspondence and requests for materials should be addressed to C.B. (email: charles-brenner@uiowa.edu).

Nicotinamide adenine dinucleotide ($NAD^+$), the central redox coenzyme in cellular metabolism[1,2] functions as a hydride group acceptor, forming NADH with concomitant oxidation of metabolites derived from carbohydrates, amino acids and fats. The $NAD^+$/NADH ratio controls the degree to which such reactions proceed in oxidative versus reductive directions. Whereas fuel oxidation reactions require $NAD^+$ as a hydride acceptor, gluconeogenesis, oxidative phosphorylation, ketogenesis, detoxification of reactive oxygen species (ROS) and lipogenesis require reduced co-factors, NADH and NADPH, as hydride donors (Fig. 1). In addition to its role as a coenzyme, $NAD^+$ is the consumed substrate of enzymes such as poly-ADPribose polymerases (PARPs), sirtuins and cyclic ADPribose synthetases[1]. In redox reactions, the biosynthetic structures of $NAD^+$, NADH, $NADP^+$ and NADPH are preserved. In contrast, PARP[3], sirtuin[4] and cyclic ADPribose synthetase[5] activities hydrolyze the linkage between the nicotinamide (Nam) and the ADPribosyl moieties of $NAD^+$ to signal DNA damage, alter gene expression, control post-translational modifications and regulate calcium signalling.

In animals, $NAD^+$-consuming activities and cell division necessitate ongoing $NAD^+$ synthesis, either through a de novo pathway that originates with tryptophan or via salvage pathways from three $NAD^+$ precursor vitamins, Nam, nicotinic acid (NA) and nicotinamide riboside (NR)[2]. Dietary $NAD^+$ precursors, which include tryptophan and the three vitamins, prevent pellagra. Though NR is present in milk[6,7], the cellular concentrations of $NAD^+$, NADH, $NADP^+$ and NADPH are much higher than those of other $NAD^+$ metabolites[8,9], such that dietary $NAD^+$ precursor vitamins are largely derived from enzymatic breakdown of $NAD^+$. Thus, although milk is a source of NR[6,7], the more abundant sources of NR, Nam and NA are unprocessed foods, in which plant and animal cellular $NAD^+$ metabolites are broken down to these compounds. Human digestion and the microbiome[10] play roles in the provision of these vitamins in ways that are not fully characterized. In addition, the conventional $NAD^+$ precursor vitamins, NA and Nam, have long been supplemented into human and animal diets to prevent pellagra and promote growth[11,12]. Though NR has been available as a GMP-produced supplement since 2013 and animal safety assessment indicates that it is as nontoxic as Nam[13], no human testing has been reported.

Different tissues maintain $NAD^+$ levels through reliance on different biosynthetic routes and precursors[14,15] (Fig. 1). Because $NAD^+$-consuming activities frequently occur as a function of cellular stresses[3] and produce Nam, the ability of a cell to salvage Nam into productive $NAD^+$ synthesis through Nam phosphoribosyltransferase (NAMPT) activity versus methylation of Nam to N-methylnicotinamide (MeNam) regulates the efficiency of $NAD^+$-dependent processes[16]. $NAD^+$ biosynthetic genes are also under circadian control[17,18]. Both NAMPT expression and $NAD^+$ levels decline in a number of tissues as a function of aging and overnutrition[19–22].

High-dose NA but not high-dose Nam is prescribed to treat and prevent dyslipidemias, although its use is limited by painful flushing[23,24]. Whereas it takes only ~15 mg per day of NA or Nam to prevent pellagra, pharmacological doses of NA can be as high as 2–4 g. Despite the >100-fold difference in effective dose between pellagra prevention and dyslipidemia treatment, we proposed that the beneficial effects of NA on plasma lipids might simply depend on function of NA as an $NAD^+$ boosting compound[1]. According to this view, sirtuin activation would likely be part of the mechanism because Nam is an $NAD^+$ precursor in most cells[14,15] but inhibits sirtuins at high doses[25].

On the basis of the ability of NR to elevate $NAD^+$ synthesis, increase sirtuin activity and extend lifespan in yeast[6,26], NR has been employed in mice to elevate $NAD^+$ metabolism and improve health in models of metabolic stress. Notably, NR allows mice to resist weight gain on high-fat diet[27], prevent noise-induced hearing loss[28] and maintain the regenerative potential of stem cells in aging mice, providing a longevity advantage[29]. In addition, the hepatic $NAD^+$ metabolome has been interrogated as a function of prediabetic and type 2 diabetic mouse models. The data indicate that levels of liver $NADP^+$ and NADPH, which are required for resistance to ROS, are severely challenged by diet-induced obesity, and that diabetes and the $NAD^+$ metabolome can be partially controlled while diabetic neuropathy can be blocked by oral NR[30].

Data indicate that NR is a mitochondrially favoured $NAD^+$ precursor[31] and in vivo activities of NR have been interpreted as depending on mitochondrial sirtuin activities[27,28], though not to the exclusion of nucleocytosolic targets[32,33]. Similarly, nicotinamide mononucleotide (NMN), the phosphorylated form of NR, has been used to treat declining $NAD^+$ in mouse models of overnutrition and aging[19,20]. Beneficial effects of NMN have been shown to depend on SIRT1[20]. However, because of the abundance of $NAD^+$-dependent processes, the effects of NR and NMN may depend on multiple targets including sirtuins, PARP family members, cADPribose synthetases, $NAD^+$-dependent oxidoreductases and NADPH-dependent ROS detoxification enzymes[30].

To translate NR technologies to people, it is necessary to determine NR oral availability and utilization in different tissues. Here we began with targeted quantitative $NAD^+$ metabolomics of blood and urine in a pilot experiment in which a healthy 52-year-old man took 1,000 mg of NR daily for 7 days. These data indicate that blood cellular $NAD^+$ rose 2.7-fold after one dose of NR and that NA adenine dinucleotide (NAAD) unexpectedly increased 45-fold. We then performed a detailed analysis of 128 mice comparing oral NR, Nam and NA in a manner that eliminated the possibility of circadian artefacts. These data indicate that NR boosts hepatic $NAD^+$ and $NAD^+$-consuming activities to a greater degree than Nam or NA. Further experiments clarified that NR is a direct precursor of NAAD and that NAAD sensitively reports on increased $NAD^+$ metabolism in mouse liver and heart. Finally, we performed a clinical study with 12 healthy human subjects at three single doses of NR. We demonstrated that NR supplementation safely increases $NAD^+$ metabolism at all doses and validated elevated NAAD as an unexpected, sensitive biomarker of boosting $NAD^+$. The unique oral bioavailability of NR in mice and people and methods established herein enable clinical translation of NR to improve wellness and treat human diseases.

## Results

**Oral NR increases human blood $NAD^+$ with elevation of NAAD.** GMP-synthesized NR showed no activity as a mutagen or toxin[13]. Despite use as an over-the-counter supplement, no data addressing human availability were available. A healthy 52-year-old male (65 kg) contributed blood and urine before seven days of orally self-administered NR (1,000 mg per morning). Blood was taken an additional nine times during the first day and at 24 h after the first and last dose. Blood was separated into peripheral blood mononuclear cells (PBMC) and plasma before quantitative $NAD^+$ metabolomics by liquid chromatography (LC)-mass spectrometry (MS)[9], which was expanded to quantify methylated and oxidized metabolites of Nam[30]. As shown in Supplementary Table 1 and Fig. 2, the PBMC $NAD^+$ metabolome was unaffected by NR for the first 2.7 h. In six measurements from time zero through 2.7 h, $NAD^+$ had a mean concentration of 18.5 µM; while Nam had a mean concentration of 4.1 µM and the methylated and oxidized

Nam metabolite, N-methyl-2-pyridone-5-carboxamide (Me2PY), had a mean concentration of 2.6 μM. However, at 4.1 h post ingestion, PBMC NAD⁺ and Me2PY increased by factors of 2.3 and 4.2, respectively.

In yeast, deletion of NR kinase 1 (NRK1) does not eliminate utilization of NR[26]. As shown in Fig. 1, NR can be phosphorylyzed to Nam by purine nucleoside phosphorylase and still contribute to NAD⁺ synthesis through Nam salvage[26,34]. However, as shown in Fig. 2, Nam concentration in the human subject's PBMCs merely fluctuated in a range of 2.6 to 7.1 μM throughout all 11 observations. The 4.2-fold increase in Me2PY concentration at the 4.1 h time point suggests that increased cellular NAD⁺ accumulation is accompanied by increased NAD⁺-consuming activities linked to increased methylation and oxidation of the Nam product.

In the subject's PBMCs at 7.7 and 8.1 h post ingestion, NAD⁺ and Me2PY peaked, increasing above baseline concentrations by 2.7-fold and 8.4-fold, respectively. At these times, unexpectedly, NAAD, the substrate of glutamine-dependent NAD⁺ synthetase[35,36], which is only expected to be produced in biosynthesis of NAD⁺ from tryptophan and NA[2], was elevated from less than 20 nM to as high as 0.91 μM. Whereas NAAD lagged the rise in PBMC NAD⁺ by one time point, the relative rise in PBMC NAD⁺ was not as pronounced as the spike in NAAD, which was at least 45-fold above the baseline level. Although contrary to expectations, these data suggested that NR might be incorporated into NAAD after formation of NAD⁺ and chased back to the NAD⁺ peak as NAD⁺ declines.

Complete NAD⁺ metabolomic data from the human subject's PBMCs, plasma and urine are provided in Figs 2–4 and Supplementary Tables 1–3. These data show that all of the phosphorylated compounds—NAMN, NAAD, NAD⁺, NADP⁺, NMN and ADPR—are found exclusively in blood cells and are not found in plasma or urine. Notably, the peak of NADP⁺, which represents cellular NADP⁺ plus NADPH oxidized in extraction, and the peak of ADPR, which signals an increase in NAD⁺-consuming activities, co-occur with peak NAD⁺. Using methods optimized for recovery of nucleotides, NR was not

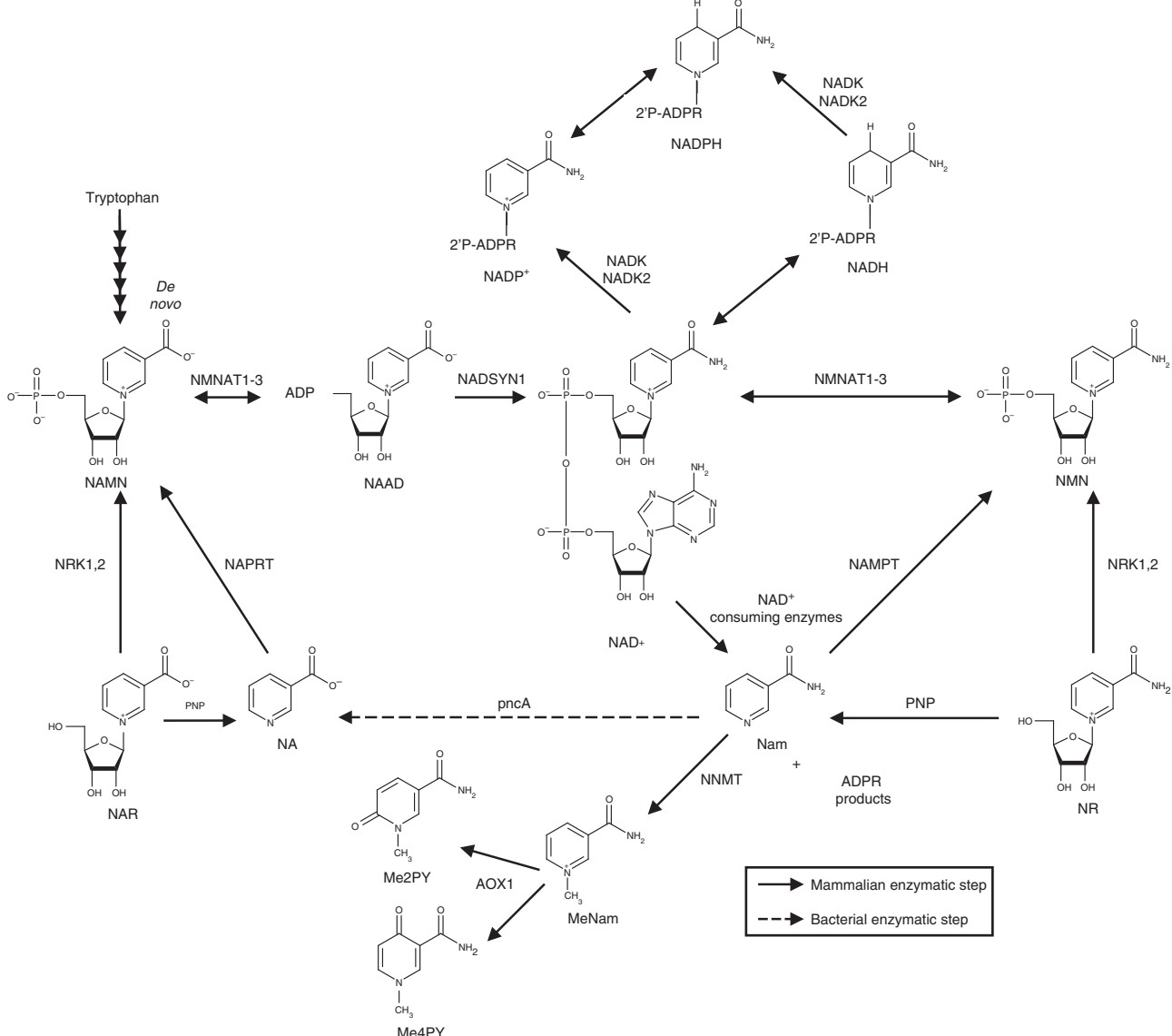

**Figure 1 | The NAD⁺ metabolome.** NAD⁺ is synthesized by salvage of the vitamin precursors, NA, Nam and NR, or from tryptophan in the *de novo* pathway. NAD⁺ can be reduced to NADH, phosphorylated to NADP⁺ or consumed to Nam. Nam can also be methylated and oxidized to waste products. NAAD was not thought to be a precursor of NAD⁺ from NR.

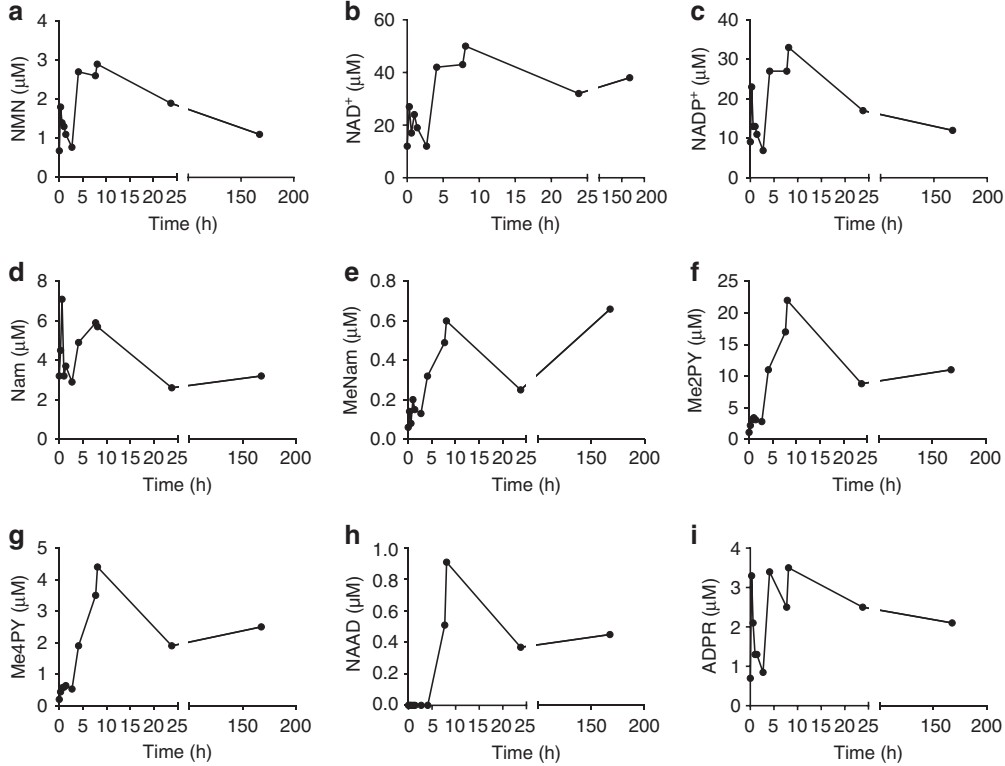

**Figure 2 | Elevation of the PBMC NAD$^+$ metabolome by oral NR in a 52 year-old male.** A healthy 52-year-old male ingested 1,000 mg NC CI daily for 1 week. PBMCs were prepared from blood collected before the first dose (0 h), at seven time points after the first dose (0.6, 1, 1.4, 2.7, 4.1, 7.7 and 8.1 h), before second dose (23.8 h) and 24 h after the seventh dose (167.6 h). Concentrations (**a**, NMN; **b**, NAD$^+$; **c**, NADP$^+$; **d**, Nam; **e**, MeNam; **f**, Me2PY; **g**, Me4PY; **h**, NAAD; **i**, ADPR) are with respect to whole blood volumes. The data indicate that the NAD$^+$ metabolome—with the exception of Nam and NAAD—is elevated by the 4.1 time point post ingestion. Whereas PBMC Nam was never elevated by NR, NAAD was elevated by the next time point. NAD$^+$ and NAAD remained elevated 24 h after the last dose.

recovered. As shown in Figs 3 and 4, the major time-dependent waste metabolite in plasma and urine was Me2PY, which rose ~10-fold from pre-dose to time points after NAD$^+$ peaked in PBMCs.

**NR is the superior hepatic NAD$^+$ precursor vitamin.** On the basis of known NAD$^+$ biosynthetic pathways[31], it was difficult to understand how levels of NAAD rose in human PBMCs after an oral dose of NR. Though NR did not elevate Nam in blood samples at any time during the $n = 1$ experiment, it remained possible that NR was partially converted to Nam before salvage synthesis to NAD$^+$. Such conversion to Nam might allow bacterial hydrolysis of Nam to NA by pncA gene products—potentially in the gut[10]—and subsequent conversion to NAD$^+$ through an NAAD intermediate. NAAD was reported in mouse liver when 500 mg kg$^{-1}$ of radioactive Nam was injected intraperitoneally (IP) into the body cavity of mice[37]. However, NAAD was observed in kidneys, ovaries, lung, heart and brain in addition to liver in mice IP-injected with 500 mg kg$^{-1}$ of NA but not Nam[38]. Moreover, careful analysis of mouse liver perfused with radioactive NA and Nam indicated that NAAD is produced from NA but not Nam at physiological concentrations[39]. To our knowledge, formation of blood or tissue NAAD from oral administration of Nam or NR has never been observed.

Although some mouse experiments have been done with IP administered NR at dosages of 1,000 mg kg$^{-1}$ twice per day[28], NR is active as an oral agent at a daily dose of 400 mg kg$^{-1}$ by supplementation into food[27,29,30] and demonstrated potent NAD$^+$ boosting activity in the $n = 1$ human experiment at

15 mg kg$^{-1}$ (Fig. 2). On the basis of weight/surface area, the conversion between human adult dose and mouse dose is a factor of 12.3 (ref. 40), suggesting that mice should be administered 185 mg kg$^{-1}$ to achieve comparable levels of supplementation with the human pilot experiment. We therefore designed a reverse translational experiment in which mice were administered 185 mg kg$^{-1}$ of NR or the mole equivalent doses of Nam and NA by oral gavage. To ascertain the timecourse by which these vitamins boost the hepatic NAD$^+$ metabolome without the complication of circadian oscillation of NAD$^+$ metabolism[17,18], we euthanized all mice at ~2 pm. Thus, gavage was performed at 0.25, 1, 2, 4, 6, 8 and 12 h before tissue harvest. To stop metabolism synchronously, mouse livers were harvested by freeze-clamping. As shown in Fig. 5, we additionally performed saline gavages at all time points and euthanized mice for quantitative NAD$^+$ metabolomic analysis to ensure that animal handling does not alter levels of NAD$^+$ metabolites. The flat timecourses of saline gavages established methodological soundness. Baseline levels of hepatic NAD$^+$ metabolites (pmol mg$^{-1}$) at 2 pm were 1,000 ± 35 for NAD$^+$, 230 ± 29 for Nam, 210 ± 20 for NADP$^+$, 66 ± 13 for ADPR and <15 for all other NAD$^+$ metabolites. Hepatic levels of NA, NAR, NAMN, NAAD have baselines of <4. As a point of orientation to quantitative metabolomics in tissue samples, 1,000 pmol mg$^{-1}$ is ~1 mM, 200 is ~200 μM and 10 is ~10 μM.

Targeted NAD$^+$ metabolomics[8,9] allows simultaneous assessment of functionally important metabolites such as NAD$^+$ and NADP$^+$ along with metabolites that could serve as biomarkers of biosynthetic processes, such as NA, NAR, NAMN, NR, NMN and NAAD. In addition, quantification of

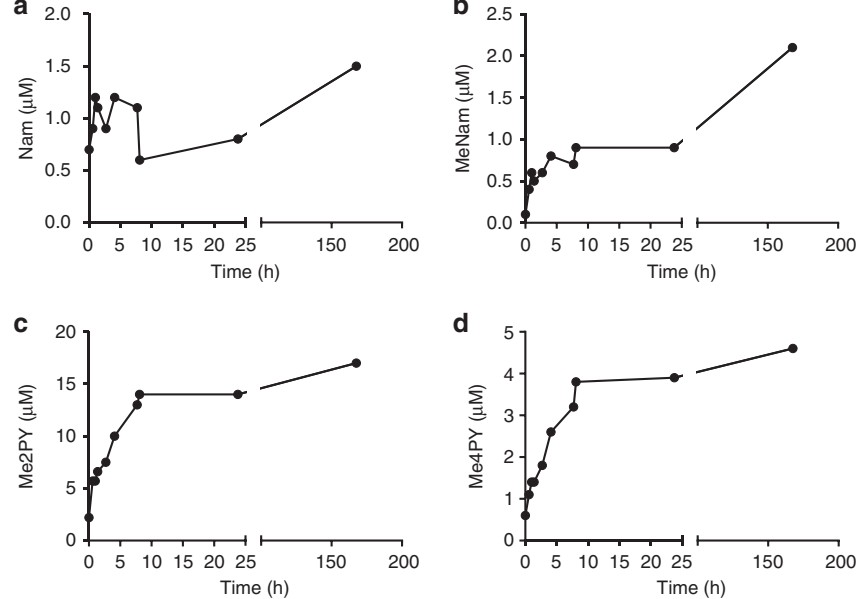

**Figure 3 | Elevation of the plasma NAD$^+$ metabolome by oral NR in a 52-year-old male.** A healthy 52-year-old male ingested 1,000 mg NC Cl daily for 1 week. Plasma samples were prepared from blood collected before the first dose (0 h), at seven time points after the first dose (0.6, 1, 1.4, 2.7, 4.1, 7.7 and 8.1 h), before second dose (23.8 h) and 24 h after the seventh dose (167.6 h). Concentrations (**a**, Nam; **b**, MeNam; **c**, Me2PY; **d**, Me4PY) are with respect to whole blood volumes. The data indicate that plasma MeNam, Me4PY and Me2PY are strongly elevated by oral NR. No phosphorylated species were found in plasma.

increases in ADPR, Nam, MeNam, Me2PY and N-methyl-4-pyridone-5-carboxamide (Me4PY) on a common absolute scale with NAD$^+$ permits assessment of increased NAD$^+$-consuming activities associated with NAD$^+$ precursor vitamin supplementation.

Hepatic concentrations of 13 NAD$^+$ metabolites were quantified in three to four mice at seven time points after gavage of saline and each vitamin. In addition, on each experimental day, three mice were gavaged with saline and euthanized to serve as time zero samples. Each vitamin produced a temporally distinct pattern of hepatic NAD$^+$ metabolites. Consistent with rapid phosphorylation of NR and NAR by NR kinases[41], the only NAD$^+$ metabolites that do not produce hepatic peaks as a function of gavage of NAD$^+$ precursor vitamins are NR and NAR (Supplementary Fig. 1a,b). The accumulation curves of some metabolites as a function of each vitamin are strikingly similar. For example, the accumulation of NMN (Fig. 5a) is nearly identical to that of NAD$^+$ (Fig. 5b) and NADP$^+$ (Fig. 5c), though at a scale of ~1:400:40, respectively. In addition, the accumulation of Me4PY (Fig. 5f) is nearly identical to that of Me2PY (Supplementary Fig. 1c).

As shown in Fig. 5b, NA produced the least increase in hepatic NAD$^+$ and also was 4–6 h faster than NR and Nam in kinetics of hepatic NAD$^+$ accumulation. When NA was provided by oral gavage, liver NA peaked (340 ± 30 pmol mg$^{-1}$) in 15 min (Fig. 5g). Hepatic NA appearance was followed by an expected peak of 220 ± 29 NAAD at 1 h post gavage (Fig. 5i) and a rise in hepatic NAD$^+$ from 990 ± 25 baseline to 2,200 ± 150 at 2 h (Fig. 5b). Hepatic NADP$^+$ due to NA (Fig. 5c) rose in parallel to that of hepatic NAD$^+$. In the hours after gavage of NA, as hepatic NAD$^+$ and NADP$^+$ fell, there was clear evidence of enhanced NAD$^+$-consuming activities with significant rises in ADPR (Fig. 5j), Nam (Fig. 5d), MeNam (Fig. 5e), Me2PY (Supplementary Fig. 1c) and Me4PY (Fig. 5f). Thus, oral administration of NA doubled hepatic NAD$^+$ from ~1 to ~2 mM through expected intermediates and produced an increase in NAD$^+$ consumption and methylated products, MeNam, Me2PY and Me4PY. Net conversion by the liver of

NA to Nam has been documented for decades[30,37,38]. Essentially, the liver transiently elevates NAD$^+$ biosynthetic capacity so long as NA is available while increasing NAD$^+$-consuming activities, thereby making Nam available to other tissues. Expression of hepatic NNMT results in net production of MeNam from NAD$^+$ precursors, which stabilizes SIRT1 protein in liver and is associated with better lipid parameters in mice and some human populations[42,43].

As shown in Fig. 5g and consistent with radioactive experiments[39], oral Nam was not used by the liver as NA because it did not produce a peak of NA at any time after gavage. Though there was a clear increase in hepatic NAD$^+$ 2 h after Nam gavage, the Nam gavage drove increased hepatic NAD$^+$ accumulation from 2 to 8 h with a peak at 8 h (Fig. 5b). Nam gavage produced two peaks of Nam in the liver (Fig. 5d), the first at 15 min, consistent with simple transport of Nam to liver. The second broad peak was coincident with elevation of NAD$^+$ and NADP$^+$ (Fig. 5c,d) and elevation of the NAD$^+$-consuming metabolomic signature of ADPR (Fig. 5j), MeNam, Me4PY and Me2PY (Fig. 5e,f and Supplementary Fig. 1c).

Of the metabolites associated with NAD$^+$-consuming activities, ADPR is the only one that must be formed from NAD$^+$ because Nam, MeNam and the oxidized forms of MeNam could appear in liver from the gavaged Nam without conversion to NAD$^+$. Interestingly, of three NAD$^+$ precursor vitamins provided in bolus at equivalent oral doses, Nam provided the least increase in ADPR (Fig. 5j). Whereas the area under the curve (AUC) of the Nam-driven rise in hepatic NAD$^+$ indicated a ~50% advantage of Nam over NA (Fig. 5b), there was a >50% deficit in Nam-driven ADPR accumulation versus NA (Fig. 5j). This is consistent with the idea that high-dose NA, though not an ideal hepatic NAD$^+$ precursor, is effective as a cholesterol agent whereas Nam is not[44] because high-dose Nam inhibits sirtuins[1]. Notably, NR is active as a cholesterol-lowering agent in overfed mice[30].

As shown in Fig. 1, Nam is expected to proceed through NMN but not NR, NAR, NaMN or NAAD en route to forming NAD$^+$. Though there was no elevation of hepatic NR or NAR with oral

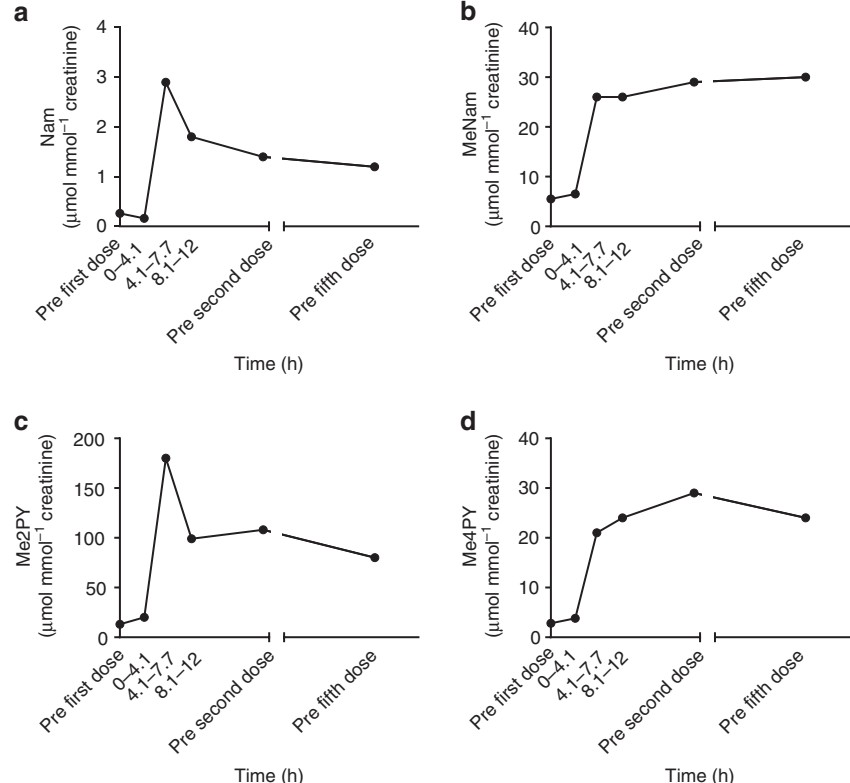

**Figure 4 | Elevation of the urinary NAD$^+$ metabolome by oral NR in a 52-year-old male.** A healthy 52-year-old male ingested 1,000 mg NC Cl daily for 1 week. Urine was collected before the first dose, in three collection fractions in the first 12 h and before the second and fifth daily dose. Concentrations (**a**, Nam; **b**, MeNam; **c**, Me2PY; **d**, Me4PY) are normalized to urinary creatinine. The data indicate that urinary MeNam, Me4PY and Me2PY are the dominant metabolites elevated by oral NR. No phosphorylated species were found in urine.

Nam, there was also little elevation of hepatic NMN—this metabolite never reached a mean value of 5 pmol mg$^{-1}$ at any time after Nam administration (Fig. 5a). Surprisingly, as shown in Fig. 5i, 2–4 h after oral Nam, NAAD was elevated to nearly 200 from a baseline of $<2$ pmol mg$^{-1}$. Elevated NAAD occurred during the broad peak of elevated hepatic NAD$^+$ and NADP$^+$ (Fig. 5b,c). These data suggest that the rise in NAAD is a biomarker of increased NAD$^+$ synthesis and does not depend on the conventionally described precursors of NAAD, namely NA and tryptophan.

As shown in Fig. 5b, NR elevated hepatic NAD$^+$ by more than fourfold with a peak at 6 h post gavage. NR also produced the greatest elevation of NMN (Fig. 5a), NADP$^+$ (Fig. 5c), Nam (Fig. 5d), NAMN (Fig. 5h), NAAD (Fig. 5i) and ADPR (Fig. 5j) in terms of peak height and AUC. Importantly, although gavage of Nam produces a peak of Nam in the liver at 15 min, the peak of Nam from NR gavage corresponds to the peak of NAD$^+$, NMN, NADP$^+$ and ADPR. These data establish that oral NR has clearly different hepatic pharmacokinetics than oral Nam. More NAD$^+$ and NADP$^+$ were produced from NR than from Nam. In addition, there was three times as much accumulation of ADPR, indicating that NR drives greater NAD$^+$-consuming activities in liver than mole equivalent doses of Nam and NA. Though it has been speculated that NR would be a more potent NAD$^+$ and sirtuin-boosting vitamin than conventional niacins[45], these are the first *in vivo* data in support of this hypothesis.

As was seen in the $n = 1$ human blood experiment, at time points in which the abundant NAD$^+$ metabolites, NAD$^+$ and NADP$^+$, were elevated by NR by $\sim$twofold or more, NAAD rose from undetectable levels to $\sim$10% of the level of NAD$^+$, thereby becoming a highly sensitive biomarker of increased NAD$^+$ metabolism. Though compounds such as MeNam, Me2PY and

Me4PY are also correlated with increased NAD$^+$ synthesis, they are waste products that can be produced without NAD$^+$ synthesis, whereas NAAD is functional NAD$^+$ precursor.

To test whether NAAD is also elevated in other tissues and through other routes of administration, we euthanized mice after 6 days of NR or saline by IP administration and analysed hepatic and cardiac NAD$^+$ metabolomes. As shown in Fig. 6, steady-state levels of hepatic NAD$^+$ and NADP$^+$ are much more responsive to NR than are steady state levels of cardiac NAD$^+$ and NADP$^+$. However, cardiac NAD$^+$ metabolism was clearly elevated on the basis of statistically significant elevation of NMN, Nam, MeNam and Me4PY. Among these metabolites, only NMN, which was elevated in the heart by approximately twofold, could be considered diagnostic for increased NAD$^+$ formation. In addition, NAMN and NAAD were increased by about $\sim$100-fold in heart and liver with NAAD rising to $\sim$10% of the concentration of heart and liver NAD$^+$ in supplemented animals. These data validate NAAD as a metabolite that sensitively and reliably marks increased NAD$^+$ metabolism even in tissues in which steady-state levels of NAD$^+$ are little changed.

**NR is incorporated into NAAD.** Appearance of hepatic NAAD after gavage of Nam or NR, and of hepatic NAMN after gavage of NR suggested that there is an NAD$^+$ and/or NMN deamidating activity when NAD$^+$ and NADP$^+$ levels are high. Alternatively, high levels of NAD$^+$ metabolites might inhibit glutamine-dependent NAD$^+$ synthetase, thereby resulting in accumulation of NAMN and NAAD derived from tryptophan. To test whether NR is incorporated into the peak of NAAD that appears after NR gavage, we synthesized NR with incorporation of deuterium at the ribosyl C2 and $^{13}$C into the carbonyl of the Nam moiety. This double-labelled NR was provided to 15 mice by oral gavage at an

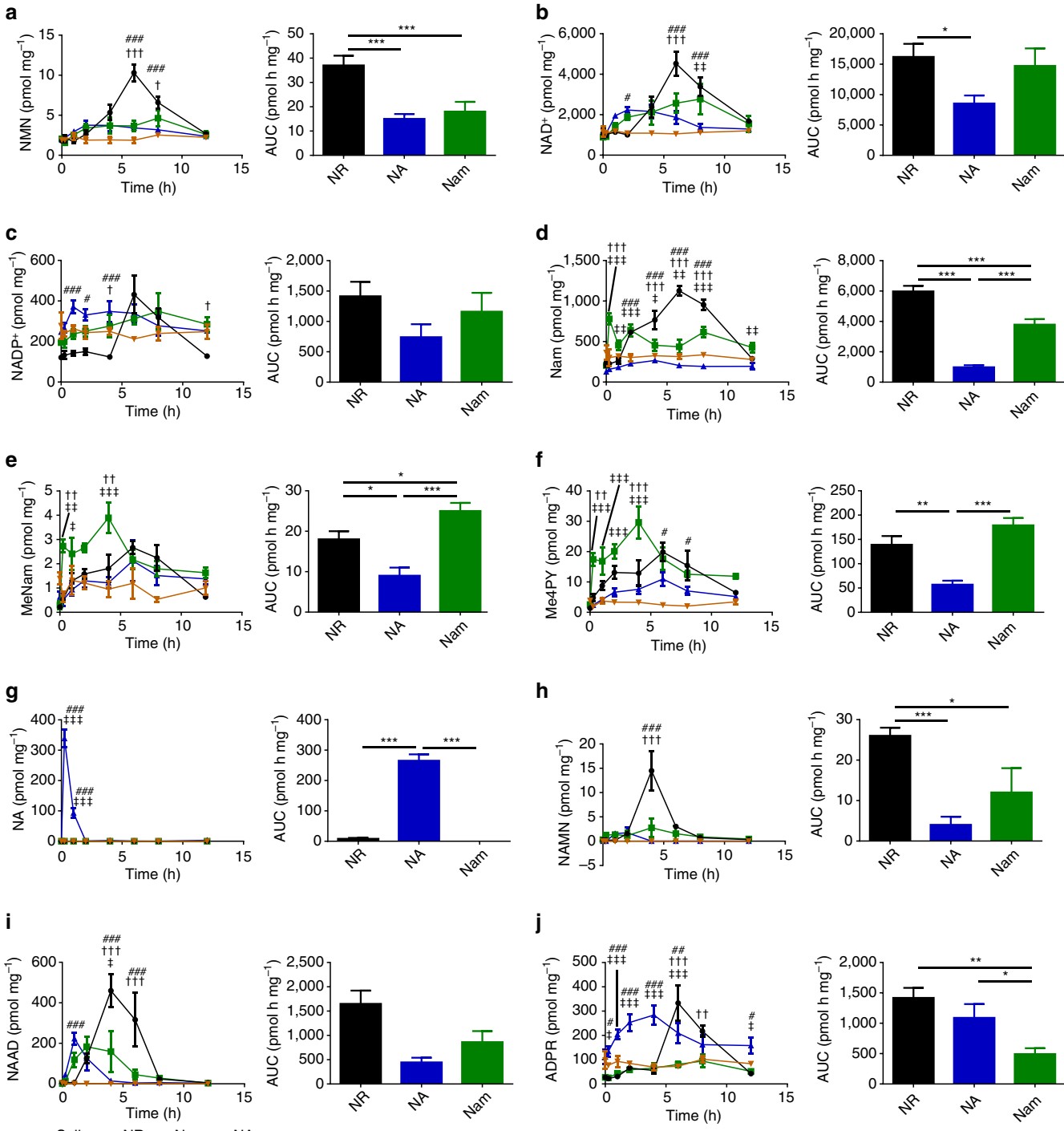

**Figure 5 | NR elevates hepatic NAD$^+$ metabolism distinctly with respect to other vitamins.** Either saline (orange, $n = 3$ per time point) or equivalent moles of NR (black, $n = 3$ per time point), NA (blue, $n = 4$ per time point) and Nam (green, $n = 4$ per time point) were administered to male C57Bl6/J mice by gavage. To control for circadian effects, gavage was performed at indicated times before a common ~2 pm tissue collection. In the left panels, the hepatic concentrations (pmol per mg of wet tissue weight) of each metabolite (**a**, NMN; **b**, NAD$^+$; **c**, NADP$^+$; **d**, Nam; **e**, MeNam; **f**, Me4PY; **g**, NA; **h**, NAMN; **i**, NAAD; and **j**, ADPR) are shown as a function of the four gavages. The excursion of each metabolite as a function of saline gavage is shown in orange; as a function of NR in black; as function of Nam in green; and NA in blue. In the right panels, the baseline-subtracted 12-hour AUCs are shown. (left) ‡$P$ value < 0.05; ‡‡$P$ value < 0.01; ‡‡‡$P$ value < 0.001 Nam versus NA; †$P$ value < 0.05; ††$P$ value < 0.01; †††$P$ value < 0.001 Nam versus NR; #$P$ value < 0.05; ##$P$ value < 0.01; ###$P$ value < 0.05 NA versus NR; (right) *$P$ value < 0.05; **$P$ value < 0.01; ***$P$ value < 0.001. The data indicate that NR produces greater increases in NAD$^+$ metabolism than Nam or NA with distinctive kinetics, that Nam is disadvantaged in stimulation of NAD$^+$-consuming activities, and that NAAD is surprisingly produced after oral NR administration.

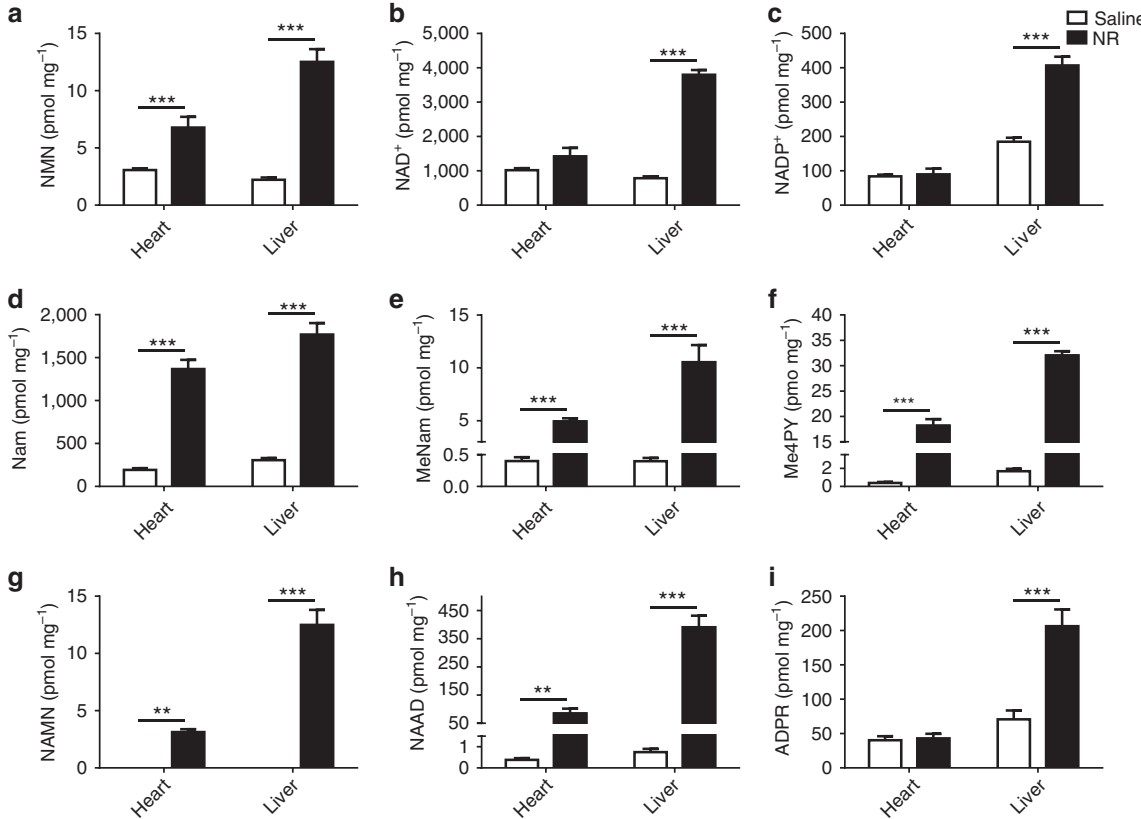

**Figure 6 | Rising NAAD is a more sensitive biomarker of elevated tissue $NAD^+$ metabolism than is $NAD^+$.** Male C57Bl6/J mice were intraperitoneally injected either saline ($n = 8$) or NR Cl (500 mg kg$^{-1}$ body weight) ($n = 6$) for 6 days. Livers and hearts were freeze-clamped and prepared for metabolomic analysis. Concentrations of $NAD^+$ metabolites (**a**, NMN; **b**, $NAD^+$; **c**, $NADP^+$; **d**, Nam; **e**, MeNam; **f**, Me4PY; **g**, NAMN; **h**, NAAD; **i**, ADPR) in heart and liver are presented in pmol mg$^{-1}$ of wet tissue weight. Data were analysed using a two-way analysis of variance followed by a Holm–Sidak multiple comparisons test. *P value < 0.05, **P value < 0.01, ***P value < 0.001. NR strikingly increased NAAD even in the heart, a tissue in which $NAD^+$ metabolism was increased without an increase in steady-state $NAD^+$ concentration.

effective dose of 185 mg kg$^{-1}$ with the same experimental design used in pharmacokinetic analysis of the three vitamins. The effect of labelled oral NR on the hepatic $NAD^+$ metabolome was first assessed at 2 h after gavage—a time point before the rise in the steady-state level of $NAD^+$ (Fig. 5b).

As shown in Fig. 7a,b, at 2 h, 54% of the $NAD^+$ and 32% of the $NADP^+$ contained at least one heavy atom while 5% of the $NAD^+$ and 6% of the $NADP^+$ incorporated both heavy atoms. Because > 50% of hepatic $NAD^+$ incorporates label before a rise in $NAD^+$ accumulation, it is clear that the $NAD^+$ pool is dynamic. As shown in Fig. 7c,d, the majority of hepatic Nam and MeNam following gavage of double-labelled NR incorporated a heavy atom, necessarily the $^{13}C$ in Nam. Because NR drives increased $NAD^+$ synthesis and ADPR production (Fig. 5), the liberated singly labelled Nam becomes incorporated into NMN and $NAD^+$ in competition with double labelled NR, thereby limiting subsequent incorporation of both labels into the $NAD^+$ pool.

Appearance of a peak of NAAD after NR administration could either be due to inhibition of *de novo* synthesis of $NAD^+$ or from a deamidating activity that occurs at high $NAD^+$. If NAAD is not derived from ingested NR, then it should not incorporate heavy atoms. However, if NAAD is derived from ingested NR, then it should incorporate heavy atoms that reflect the rate at which the putative deamidating activity occurs with respect to $NAD^+$-consuming activities and the degree of heavy atom incorporation into $NAD^+$. As shown in Fig. 7e, at the 2 h time point, NAAD contained roughly the same heavy atom

composition as $NAD^+$ (Fig. 7a), that is, 45% contained at least one heavy atom and 8% incorporated both heavy atoms. Thus, NR is the biosynthetic precursor of $NAD^+$, $NADP^+$ and NAAD. The data suggest that the activity that converts $NAD^+$ to NAAD occurs at high $NAD^+$ concentrations at a rate comparable to the rate of $NAD^+$ turnover to Nam. Given the lack of formation of NA from either Nam or NR in the mouse liver (Fig. 5), the only other reasonable possibility is that NMN is deamidated to NAMN when $NAD^+$ metabolism increases. Incorporation of the Nam and ribosyl moieties of NR into NAAD establishes this metabolite as both a biomarker of increased $NAD^+$ metabolism and a direct product of NR utilization.

**NR safely increases PBMC $NAD^+$ metabolism and NAAD in people.** The $n = 1$ human experiment illustrated the potential of 1,000 mg NR to boost human $NAD^+$ metabolism. We therefore conducted a controlled experiment with 12 consented healthy men and women to determine the effect of three single doses of NR on blood and urine $NAD^+$ metabolites with monitoring of subjects for potential adverse events. Considering that the recommended daily allowance of vitamin B3 as Nam or NA is $\sim 15$ mg per adult, we tested three doses of the higher molecular weight compound NR Cl (100, 300 and 1,000 mg) that correspond to 2.8, 8.4 and 28 times recommended daily allowance. Body weights of the subjects varied. However, we had already observed the timecourse of changes in a human $NAD^+$

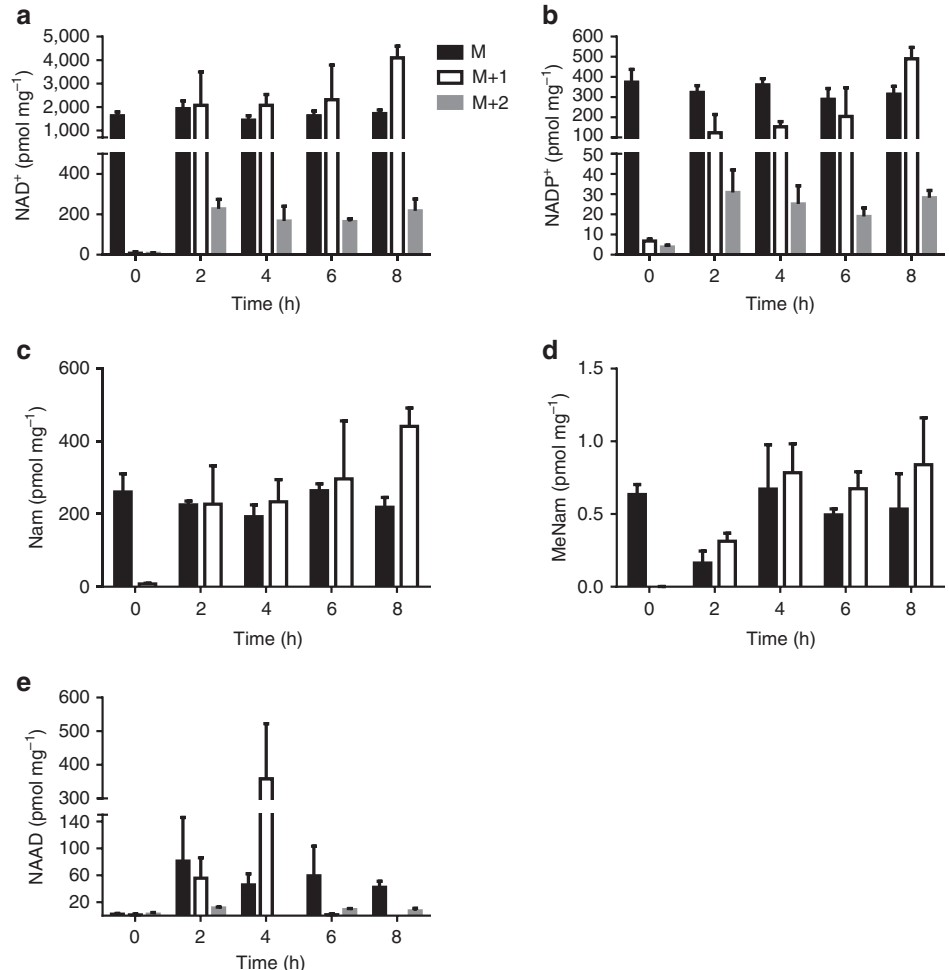

**Figure 7 | NR contributes directly to hepatic NAAD.** Double-labelled NR was orally administered to mice. At indicated times after gavage, mice ($n = 3$) were euthanized and livers freeze-clamped for isotopic enrichment analysis. Data are plotted as pmol mg$^{-1}$ of wet tissue weight. At the 2 h time point, hepatic NAD$^+$ (**a**) shows 54% incorporation of M+1 and M+2 species, indicating that more than half of the NAD$^+$ is turned over before there is a rise in steady-state NAD$^+$. At 2 h, hepatic NADP$^+$ (**b**) shows 32% incorporation of label(s) from ingested NR. At all time points, half or more of hepatic Nam (**c**) and MeNam (**d**) carries an NR-derived label, indicating that NR has driven rapid NAD$^+$ synthesis and consumption. At 2 h after gavage, 45% of hepatic NAAD (**e**) incorporates NR-derived label(s). At 2 h, hepatic NAD$^+$ (**a**), NADP$^+$ (**b**) and NAAD (**e**) pools incorporate 5, 6 and 8% of double-labelled NR indicating that NR is a direct precursor of all three metabolites and excluding the possibility that the NR-driven increase in NAAD is due to inhibition of *de novo* synthesis.

metabolome with daily doses of 1,000 mg in a healthy 65 kg male. Participants were randomized to receive doses of NR in different sequences with 7-day washout periods between data collection. Participants and investigators were blinded to doses. Blood and urine collections were performed over 24 h following each dose. Participants were asked to self-report perceived discomforts.

At 500 mg of niacin, 33 of 33 participants experienced flushing compared with one out of 35 participants on placebo[46]. In this study, two individuals self-reported flushing at the 300 mg dose but not at the 100 mg or 1,000 mg dose, and two individuals self-reported feeling hot at the 1,000 mg dose but not at lower doses. Over the total of 36 days of observation of study participants, there were no serious adverse events and no events that were dose-dependent. To assess whether NR might be associated with authentic and dose-dependent episodes of flushing, future experiments will incorporate a validated flushing symptom questionnaire[47].

As shown in Fig. 8 and Supplementary Data Files 1 and 2, the NAD$^+$ metabolome was quantified in the PBMC and plasma fractions at pre-dose and at 1, 2, 4, 8 and 24 h after receiving oral NR. Urinary NAD$^+$ metabolites (Supplementary Data File 3)

were quantified in pre-dose, 0–6 h, 6–12 h and 12–24 h collections.

As shown in Fig. 5, inbred, chow-fed male mice supplemented with NAD$^+$ precursor vitamins by gavage and euthanized at $\sim 2$ pm produced hepatic NAD$^+$ metabolomic data with little variation. However, blood samples from people exhibited greater variation, due to differing baseline levels of metabolites and variable pharmacokinetics, both of which are likely due to genetic and nutritional changes between subjects (Fig. 8). In PBMCs, eight key metabolites were quantified in at least 10 subjects at all time points at each dose. For each metabolite, we plotted average concentration as a function of dose and time, calculated whether NR elevated that metabolite, plotted the averaged peak concentration of the metabolite as a function of dose, and calculated the dose-dependent AUC of the metabolite attributable to NR supplementation. Unlike a drug metabolism study in which metabolites appear only after administration, most NAD$^+$ metabolites are present before supplementation, such that the AUC attributable to supplementation is a time-zero baseline-subtracted AUC. Thus, it is possible to calculate not only the AUC rise in metabolites but also the per cent increase in

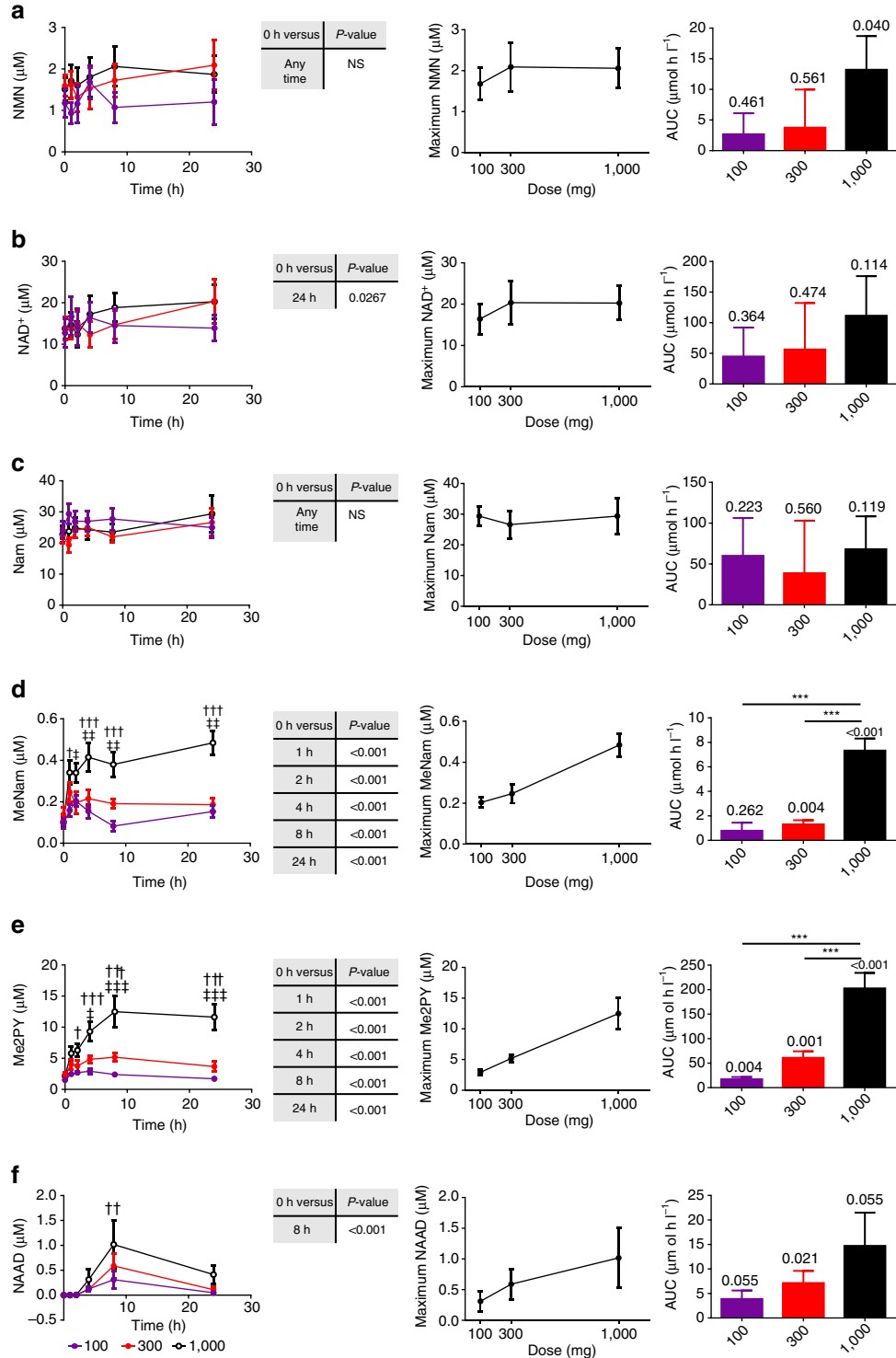

**Figure 8 | Dose-dependent effects of NR on the NAD$^+$ metabolome of human subjects.** Time-dependent PBMC NAD$^+$ metabolomes from $n = 12$ healthy human subjects were quantified after three different oral doses of NR. In each left panel, the blood concentration of a metabolite as a function of dose and time is displayed (**a**, NMN; **b**, NAD$^+$; **c**, Nam; **d**, MeNam; **e**, Me2PY; **f**, NAAD). [#]P value < 0.05; [##]P value < 0.01 100 mg versus. 300 mg; [††]P value < 0.01; [†††]P value < 0.001 100 versus. 1,000; [‡]P value < 0.05; [‡‡]P value < 0.01 300 versus. 1,000. A Dunnett's test was performed comparing the average concentration of each metabolite at each time point to the concentration of that metabolite at time zero. Significant elevations of NAD$^+$, MeNam, Me2PY and NAAD are indicated. In each middle panel, the averaged maximum metabolite concentration per dose is plotted. In each right panel, the background-subtracted metabolite AUCs are displayed with a one sample *t*-test comparing the AUC to background above each bar. In addition, asterisks indicate dose-dependent increases in metabolite AUC (*P value < 0.05; **P value < 0.01; ***P value < 0.001). The data indicate that all doses of NR elevated 8 h NAAD and 24 h NAD$^+$, and that additional NAD$^+$ metabolites were elevated dose dependently with statistical significance by multiple comparisons.

AUC in these metabolites attributable to dose-dependent NR supplementation.

Collapsing the data into pre-dose versus 24 h levels of each metabolite at all doses, NR significantly elevated PBMC $NAD^+$ (Fig. 8b), MeNam (Fig. 8d) and Me2PY (Fig. 8e) and significantly elevated PBMC NAAD (Fig. 8f) at 8 h. In contrast, NR did not produce a statistically significant all-dose elevation of NMN (Fig. 8a) or Nam (Fig. 8c) at any time point.

The averaged peak concentration of MeNam (Fig. 8d), Me2PY (Fig. 8e) and NAAD (Fig. 8f) increased monotonically with increased NR doses. Of these metabolites, only NAAD was below the detection limit in individuals before they took NR, qualifying this metabolite as a biomarker of supplementation. Nam (Fig. 8c) exhibited no tendency towards higher cellular concentrations with higher doses of NR. NMN tended to rise (Fig. 8a) and $NAD^+$ rose (Fig. 8b) to higher concentrations of $\sim 2$ and 20 $\mu$M, respectively, in people taking 300 and 1,000 mg doses of NR versus people taking 100 mg doses. Thus, 100 mg supplementation produced an average $\sim 4 \pm 2 \mu$M increase in PBMC $NAD^+$, whereas the higher doses produced average $\sim 6.5 \pm 3.5 \mu$M increases in PBMC $NAD^+$. No sex differences were discovered.

As was first seen in the $n = 1$ human experiment and in mouse liver experiments, NAAD is the most sensitive biomarker of effective $NAD^+$ supplementation because it is undetectable in the blood of people before supplementation. At all doses, the peak shape of NAAD indicated that $NAD^+$ metabolism is most greatly boosted at 8 h with significant supplementation at 4 h and significant supplementation remaining at 24 h. At the 8 h peak, the average concentration of NAAD was elevated to $0.56 \pm 0.26$, $0.74 \pm 0.27$ and $1.24 \pm 0.51 \mu$M in PBMCs from volunteers taking 100, 300 and 1,000 mg single doses of NR, respectively.

We plotted pre-dose-subtracted AUCs of each metabolite as a function of dose of NR. With the exception of Nam, the levels of which were unaffected by NR, NR produced or tended to produce dose-dependent elevation of the entire $NAD^+$ metabolome (Fig. 8). In plasma, levels of meNam, me2PY and me4PY also rose in a dose-dependent manner and were identified at concentrations similar to those in the PBMC fraction. The methylated and oxidized Nam derivatives were accompanied by low levels of NAR, which increased with higher doses of NR. Urinary metabolites were similar to plasma metabolites.

In Supplementary Table 4, the average 24 h baseline-subtracted AUC of each metabolite is expressed as a percentage increase in that metabolite at each dose of NR. Once again, Nam and NMN showed essentially no increase in blood cell concentrations with respect to baseline concentrations: averaged AUCs never rose by 50% above baseline. However, $NAD^+$, meNam, me2PY and NAAD rose or tended to rise in dose-dependent manners. The effect size of the rise in NAAD ($\sim 2,900\%$) was much greater than the effect sizes of the rise in me2PY ($\sim 600\%$), meNam ($\sim 200\%$) or $NAD^+$ ($\sim 90\%$). AUC increases of NAAD, me2PY and meNam achieved statistical significance with respect to lower doses of NR.

## Discussion

Despite $>75$ years of human use of NA and Nam[11] and $>10$ years of preclinical NR research[6], there has never been a quantitative metabolomic or pharmacokinetic comparison of the three $NAD^+$ precursor vitamins in any system. In terms of elevation of mouse liver $NAD^+$, we discovered that NR is more orally bioavailable than Nam, which is more orally bioavailable than NA (Fig. 5b). The three precursors also differ in the degree to which they promote accumulation of ADPR, a measure of sirtuin and other $NAD^+$-consuming activities. As shown in Fig. 5j, the ability of NR to elevate ADPR exceeded that of Nam by $\sim 3$-fold. This validates NR as the favoured $NAD^+$ precursor

vitamin for increasing $NAD^+$ and $NAD^+$-consuming activities in liver.

NR, Nam and NA each have unique pharmacokinetic profiles in mouse liver, both in terms of kinetics of $NAD^+$ formation and the population of $NAD^+$ metabolites as a function of time. As shown in Fig. 5d, Nam is the only vitamin precursor of $NAD^+$ that produces elevated hepatic Nam 15 min after oral administration and, as shown in Fig. 5g, NA is the only precursor that produces elevated NA 15 min after oral administration. These data exclude the possibility that all three vitamins are utilized through the Preiss-Handler pathway in liver or that oral NR is used exclusively as Nam. Additionally, we have demonstrated that utilization of both NR and extracellular NMN are limited by activity of the NR kinase pathway[48].

When PBMCs were analysed from the first person to ingest NR, NAAD was observed to increase at least 45-fold from a baseline of less than 20 nM to a peak value of nearly 1 $\mu$M. This occurred concomitant with a rise in $NAD^+$ from $\sim 18.5$ to 50 $\mu$M. NAAD was also observed to be elevated in liver when mice were orally administered $NAD^+$ precursor vitamins. In addition, NR led to striking elevation of NAAD in the heart, a tissue that increases $NAD^+$ metabolism without increasing steady-state $NAD^+$.

Surprisingly, NA, the only precursor expected to proceed to $NAD^+$ through an NAAD intermediate, produced the least NAAD. Indeed, although Nam and NR never produced peaks of hepatic NA or NAR, both produced peaks of hepatic NAAD during the periods in which these compounds elevated hepatic $NAD^+$. The temporal basis of the NAAD excursions suggested that elevating $NAD^+$ (Fig. 5b) not only stimulates accumulation of $NAD^+$-consumption products ADPR (Fig. 5j), Nam (Fig. 5d), MeNam (Fig. 5e) and Me4PY (Fig. 5f), but also stimulates retrograde production of NAAD (Fig. 5i) and NAMN (Fig. 5h). According to this view, as the rate of $NAD^+$ synthesis increases, a previously unknown activity would deamidate $NAD^+$ to NAAD. Alternatively, similar conditions could result in NMN deamidation, giving rise to NAMN and NAAD.

In mouse liver, the apparent flux through this pathway is quite significant: the NR-driven peak of NAAD amounted to 10% of the NR-attributable peak of $NAD^+$. Production of high levels of NAAD from $NAD^+$ could therefore account for the NR-stimulated peak in NAMN because NAMN adenylyltransferase is a reversible enzyme[49]. Striking elevation ($\sim 100$-fold) of NAAD was also seen in the heart of mice supplemented with NR after 6 days of IP administration (Fig. 6).

The hypothesis that NAAD is formed from NR in vivo was tested by administering NR labelled in the Nam and ribosyl moieties. As shown in Fig. 7, NR stimulates appearance of double-labelled NAAD (8% of total) at the same time in which 5% of $NAD^+$ is double-labelled. The biochemical basis for a potential $NAD^+$ deamidation reaction is unknown. However, glutamine-dependent $NAD^+$ synthetase is irreversible[35,36]. One intriguing possibility is that NAAD is formed by the long-sought enzyme that forms intracellular NAADP[50]. According to this view, an NADP deamidase may be responsible for formation of NAADP—this same activity might deamidate $NAD^+$ at high concentrations forming NAAD. Unlike ADPR and methylated Nam waste products, NAAD is not only a biomarker of elevated $NAD^+$ metabolism but also a reserve metabolite that contributes to elevated $NAD^+$ over time.

Finally, in the first clinical study of NR, we established that blood $NAD^+$ metabolism is increased by single 100, 300 and 1,000 mg doses of NR without dose-dependent increases in PBMC Nam or serious adverse events.

In people, as in mice, NAAD is the most sensitive biomarker of boosting $NAD^+$. While 1,000 mg of NR elevated PBMC $NAD^+$

from ∼12 to ∼18 µM and generated a ∼90% increase in 24 h AUC, NAAD was elevated from below the limit of quantification to ∼1 µM and generated a 2,900% increase in 24 h AUC. The ability to detect NAAD in human samples is expected to aid conduct of clinical testing of NR. Availability of over-the-counter supplements can complicate clinical trials because patients may enrol to obtain compounds they expect to bring benefits and be inclined to take supplements in case they are assigned to placebo. Detection of NAAD should therefore be incorporated in phase II and III trials to eliminate the confounding effects of off-study NR use.

## Methods

**Materials and reagents.** NR Cl was produced under GMP conditions. Me2PY and Me4PY were purchased from TLC PharmaChem Inc. (Vaughan, Ontario, Canada). All other unlabelled analytes were purchased from Sigma-Aldrich (St Louis, MO) at highest purity. Internal standards $[^{18}O_1]$- Nam and $[^{18}O_1]$-NR were prepared as described[51,52]. $[^{18}O_1$-$D_3]$-MeNam was prepared by alkylation of $[^{18}O_1]$-Nam with deuterated iodomethane. $^{13}$C-NA and $[D_4]$-NA were purchased from Toronto Chemical Research (Toronto, Ontario, Canada) and C/D/N Isotopes, Inc. (Pointe-Claire, Quebec, Canada), respectively. To prepare $[^{13}C, D_1]$-NR, we first converted $^{13}$C-NA to $^{13}$C-Nam (ref. 53) and D-[2-$D_1$]-ribose (Omicron Biochemicals, South Bend, IN) to the labelled D-ribofuranose tetraacetate[54]. The labelled D-ribofuranose-tetraacetate and Nam were then used to synthesize double-labelled NR[55]. $[^{13}C]$-labelled nucleotides standards were prepared by growing yeast in U-$^{13}$C-glucose and extracting as described[9].

**Pilot human experiment.** After overnight fasting, a healthy 52-year-old male self-administered 1,000 mg of NR Cl orally at 8 am on 7 consecutive days. Blood and urine were collected for quantitative NAD$^+$ metabolomic analysis. The participant took 0.25 g of NA to assess sensitivity to flushing and self-reported painful flushing that lasted 1 h. No flushing was experienced on NR. The study was submitted for approval by the University of Iowa institutional review board (IRB), which ruled it not subject to human subjects research on the basis of informed self-administration[56].

**Mice.** For gavage experiments, 12-week-old male C57Bl/6J mice (Jackson Laboratories, Bar Harbour, ME) were housed 3–5 mice per cage on a chow diet (Teklad 7013) for one week before the experiment. Body weight-matched groups were randomly assigned to be given either 185 mg NR Cl per kg body weight ($n=3$) or equimole amounts of NA ($n=4$) or Nam ($n=4$) by saline gavage. On each experimental day, a saline injection ($n=3$) was performed and served as time point zero and an additional saline gavage ($n=3$) timecourse was performed. To avoid circadian effects, timecourses were established such that all tissue harvests were performed at ∼2 pm. Double-labelled NR ($n=3$) was also administered by gavage against a saline control ($n=3$). With protocols approved by the University of Iowa Office of Animal Resources, mice were live-decapitated and the medullary lobe of the liver was freeze-clamped at liquid nitrogen temperature. For IP experiments, 6–8-week-old, male C57Bl/6J mice were injected with either PBS or 500 mg NR Cl per kg body weight for 6 days. With protocols approved by the Institutional Animal Care and Use Committee of University of Utah, mice were anaesthetised by chloral hydrate and livers and hearts were freeze-clamped at liquid nitrogen temperature. Tissues were stored at −80 °C before analysis.

**Clinical trial.** A randomized, double-blind, three-arm crossover pharmacokinetic study of oral NR chloride was performed at 100, 300 and 1,000 mg doses (Clinicaltrials.gov Identifier NCT02191462). Twelve healthy, non-pregnant subjects (six male and six female) between the ages of 30 and 55 with body mass indices of 18.5–29.9 kg m$^{-2}$ were recruited and randomized to one of three treatment sequences after screening, passing eligibility criteria and providing informed consent. Overnight fasted subjects received a single morning dose of either 100 mg, 300 mg, or 1,000 mg of NR on 3 test days separated by 7-day periods in which no supplement was given. To evaluate pharmacokinetics, blood was collected and separated into plasma and PBMC fractions for analysis of the NAD$^+$ metabolome at pre-dose and again at 1, 2, 4, 8 and 24 h. Urine was collected pre-dose and in 0–6 h, 6–12 h and 12–24 h fractions. Safety, vitals, biometrics, complete blood counts and a comprehensive metabolic panel were assessed at time zero and 24 h after each dose. The study was reviewed and approved by the Natural Health Products Directorate, Health Canada and IRB Services, Aurora, Ontario. Written informed consent was obtained from each subject at the screening visit before all study-related activities.

Exclusion criteria: women who were pregnant, breastfeeding or planning to become pregnant during the course of the trial; use of natural health products/dietary supplements within 7 days before randomization and during the course of the study; use of vitamins or St John's Wort 30 days before study enrolment; use of supplements containing NR within 7 days before randomization and the

course of the study; use of nutritional yeast, whey proteins, energy drinks, grapefruit and grapefruit juice, dairy products, alcohol for 7 days before the study; consumption of >2 standard alcoholic drinks per day or drug abuse within the past 6 months; smoking; blood pressure ≥140/90; use of blood pressure medications; use of cholesterol-lowering medications; metabolic diseases or chronic diseases; use of acute over-the-counter medication within 72 h of test product dosing; unstable medical conditions as determined by the qualified investigator; immune compromised conditions including organ transplantation or human immunodeficiency virus; clinically significant abnormal lab results at screening (for example, aspartate transaminase and/or alanine transaminase >2× upper limit of normal (ULN), and/or bilirubin >2× ULN); planned surgery during the course of the trial; history of or current diagnosis of any cancer (except successfully treated basal cell carcinoma or cancer in full remission >5 years after diagnosis); history of blood/bleeding disorders; blood donation in the previous 2 months; participation in a clinical research trial within 30 days before randomization; allergy or sensitivity to study supplement ingredients or to any food or beverage provided during the study; cognitive impairment and/or inability to give informed consent; any other condition which in the qualified investigator's opinion may have adversely affected the subject's ability to complete the study or its measures or which may have posed significant risk to the subject.

**Sample preparation and targeted quantitative metabolomics.** Dual extractions were carried out for complete analysis of the NAD$^+$ metabolome. For analysis of NR, Nam, NA, MeNam, Me2PY and Me4PY (group A analytes), samples were spiked with 60 pmol of $[^{18}O_1]$-Nam, $[^{18}O_1]$-NR and $[D_3, ^{18}O_1]$-MeNam and 240 pmol $[D_4]$-NA (internal standard (IS) A). For analysis of NAD$^+$, NADP$^+$, NMN, NAR, NAMN, NAAD and ADPR (group B analytes), samples were dosed with $^{13}$C-yeast extract (IS B) as described[9].

*Human samples.* One hundred microlitre of urine was mixed with 20 µl of IS A in 5% (v/v) formic acid or IS B in water for the analysis of group A and B analytes, respectively. Fifty microlitre of ice-cold methanol was added and the mixture vortexed before centrifugation at 16.1kg at 4 °C for 10 min. Supernatants were injected without further dilution and analysed as described below. Standard curves and quality controls for the complete analysis were prepared in the same manner as described for urine samples but in water.

To quantify group A analytes in plasma, 100 µl of plasma was added to 20 µl of IS A prepared in 5% (v/v) formic acid and mixed with 400 µl of ice-cold methanol. The mixture was allowed to sit on ice for 20 min then centrifuged as described for urine. After drying under vacuum overnight at 35 °C, the sample was reconstituted in 100 µl of water. To quantify group B analytes, 100 µl of plasma was added to 10 µl of IS B in water and mixed with 300 µl of acetonitrile with vortexing for 15 s. After briefly resting on ice, the samples were centrifuged as above. Supernatants were applied to Phenomenex Phree SPE cartridges (Torrance, CA) and the flow-through collected. 200 µl of aqueous acetonitrile (four volumes acetonitrile: one volume water) was also applied and the flow-through collected. The flow-through from both steps was combined and dried via speed vacuum. Samples were reconstituted in 60 µl of water. Standard curves and quality controls for both analyses were prepared in donor plasma (University of Iowa DeGowin Blood Center, Iowa City, IA) and extracted using the same method employed for plasma samples.

PBMC fractions were thawed on ice and simultaneously extracted for both A and B analyses when possible. 100 µl of sample was added to either 20 µl of IS A in 5% formic acid (v/v) or 10 µl IS B in water for quantification of group A and B analytes, respectively. Samples were then mixed with 300 µl of acetonitrile and vortexed for 15 s. Samples were shaken for 5 min at 40 °C then centrifuged as described above. For group A analytes, supernatants were dried via speed vacuum overnight at 35 °C after this step. For group B analytes, supernatant was applied to Phenomenex Phree SPE cartridges and treated in the same manner as described above for quantification of group B analytes in plasma. Immediately before analysis, samples were reconstituted in either 100 µl of 10 mM ammonium acetate with 0.1% formic acid (for group A analyte quantification) or 100 µl of 5% (v/v) aqueous methanol (for group B analyte quantification). Standard curves were prepared in water and processed in the same manner as samples.

*Murine samples.* For the IP experiment, male C57Bl/6J mice were injected with either saline or NR Cl (500 mg kg$^{-1}$) for 6 days. On the day of tissue harvests, mice were anaesthetized by chloral hydrate before collection of liver and heart both of which were freeze-clamped in liquid nitrogen. All tissues were stored at −80 °C before extraction.

Murine liver and heart obtained by freeze-clamp were pulverized using a Bessman pulverizer (100–1,000 mg size) (Spectrum Laboratories, Rancho Dominguez, CA) cooled to liquid N$_2$ temperatures. Each pulverized liver and heart sample was aliquoted (5–20 mg) into two liquid N$_2$ cooled 1.5 ml centrifuge tubes, which were stored at −80 °C until analysis. Before extraction, murine liver sample identities were masked and sample order randomized.

Before extraction, IS A and IS B were added to separate aliquots resting on dry ice for quantification of group A and B analytes, respectively. Samples were extracted by addition of 0.1 ml of buffered ethanol (three volumes ethanol: one volume 10 mM HEPES, pH 7.1) at 80 °C. Samples were vortexed vigorously until thawed, sonicated in a bath sonicator (10 s followed by 15 s on ice, repeated twice for liver and thrice for heart), vortexed, then placed into a Thermomixer (Eppendorf, Hamburg, Germany) set to 80 °C and shaken at 1,050 r.p.m. for 5 min.

Samples were centrifuged as described above. Clarified supernatants were transferred to fresh 1.5 ml tubes and dried via speed vacuum for two h. Before LC-MS analysis, samples were resuspended in 40 µl of 10 mM ammonium acetate (>99% pure) in LC-MS-grade water. Sample preparation following $[^{13}C_1, D_1]$-NR administration differed only in the following respect. In lieu of IS A, 60 pmol of $[D_4]$-Nam and $[D_3, {}^{18}O_1]$-MeNam and 240 pmol of $[D_4]$-NA were added to sample. Standard curves were prepared in water without extraction.

*LC-MS.* Separation and quantitation of analytes were performed with a Waters Acquity LC interfaced with a Waters TQD mass spectrometer operated in positive ion multiple reaction monitoring mode as described[9]. MeNam, Me2PY and Me4PY were added to the analysis and detected using the following transitions: MeNam ($m/z$ 137 > 94, cone voltage = 8 V, collision energy = 20 V); Me2PY ($m/z$ 153 > 107, cone voltage = 44 V, collision energy = 22 V); and Me4PY ($m/z$ 153 > 136, cone voltage = 24 V, collision energy = 14 V). For the analysis of urine, plasma and murine liver, heart and blood cells, group A analytes were separated as described for the acid separation[9]. In blood cells, group A analytes were separated on a 2.1 × 150 mm Synergy Fusion-RP (Phenomenex, Torrance, CA) using the same gradient and mobile phase as described for the acid separation[9]. For human samples, group B analytes were separated using the mobile phase and gradient as previously described for the alkaline separation[9]. Murine liver and heart extracts were analysed using a slightly altered alkaline separation on a 2.1 × 100 mm Thermo Hypercarb column. Specifically, flow rate was increased to 0.55 ml min$^{-1}$ and run time shortened to 11.6 min. Separation was performed using a modified gradient with initial equilibration at 3% B, a 0.9 min hold, a gradient to 50% B over 6.3 min, followed by a 1 min wash at 90% B and a 3 min re-equilibration at 3% B.

Analytes in human plasma were quantified by dividing their peak areas by IS peak areas and comparing the ratio to a background-subtracted standard curve. Analytes in all other matrices were quantified by dividing their peak areas by IS peak areas and comparing the ratio to a standard curve in water. Urinary metabolites were normalized to creatinine concentrations. Hepatic metabolites were normalized to wet liver weights.

For both human and mouse samples, samples were transferred to Waters polypropylene plastic total recovery vials (Part # 186002639) after extraction or preparation and stored in a Waters Acquity H class autosampler maintained at 8 °C until injection. In all cases, 10 µl of extract was loaded onto the column.

For analysis of enrichment in murine liver, metabolites were separated following the same LC procedure described above and detected using a Waters Premier Q-TOF operated in positive ion, full-scan mode. Leucine enkephalin was infused to ensure high mass accuracy. Enrichment data were corrected for natural isotope abundance based on theoretical isotope distribution, 13-carbon abundance skew and the purity of the labelled standard (3/97% $[^{13}C_1]$-NR/$[^{13}C_1, D_1]$-NR). Quantification was performed on the Waters TQD as described above and used to determine the quantity of non-labelled and labelled metabolites. Separation and quantification of analytes was performed with a Waters Acquity LC interfaced with a Waters TQD mass spectrometer operated in positive ion multiple reaction monitoring mode. Enrichment analysis was performed with a Waters Q-TOF Premier mass spectrometer operated in positive ion, full-scan mode with the same LC conditions as described for non-enrichment experiments.

**Statistical analyses.** Statistical analyses were performed in GraphPad Prism version 6.00 for Windows, (La Jolla, CA). Sample sizes were sufficiently powered ($1-\beta = 0.8$, $\alpha = 0.05$) to detect at least a 2-fold difference in NAD$^+$ concentration. Murine liver data were analysed using two-way analysis of variance, whereas human blood cells were analysed using a repeated measure, two-way analysis of variance. Holm–Sidak and Tukey's multiple comparisons tests were performed when comparing more than two groups. AUCs in blood cells were calculated after subtracting pre-dose metabolite concentrations of each experimental series. For mouse data, AUCs were calculated as described[57] after subtracting the saline group for that day and propagating error. Data are expressed as means ± s.e.m. Group variances were similar in all cases. A *P* value <0.05 was considered significant.

**Data availability.** Metabolomic data have been deposited in Metabolights (https://www.ebi.ac.uk/metabolights/) under accession code MTBLS368.

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

## Acknowledgements

This study was supported by grants from the Roy J. Carver Trust, ChromaDex and National Institutes of Health (R21-AA022371) to C.B., from the National Institutes of Health (R01-HL108379) to E.D.A. and from the Biotechnology & Biological Sciences Research Council (BB/N001842/1) to M.E.M. We thank Dale Wilson and staff at KGK Synergize, Inc. (London, Ontario, Canada) for conducting the $n = 12$ clinical study.

## Author contributions

S.A.J.T., B.J.W, F.J., R.W.D., M.E.M. and C.B. designed the experiments. S.A.J.T., M.S.S., B.J.W. and Z.L. performed the experiments and analysed the data with F.J., R.W.D., E.D.A., M.E.M. and C.B. P.R. performed the synthesis under direction of M.E.M. S.A.J.T. and C.B. wrote the manuscript. All authors edited the manuscript and figures.

## Additional information

**Competing financial interests:** F.J., R.W.D. and C.B. own stock in ChromaDex, the supplier of NR and sponsor of the clinical study. F.J. and R.W.D. are employees of ChromaDex. M.E.M. has received research grants and serves as a consultant for ChromaDex. C.B. has received a research grant and serves on the scientific advisory board of ChromaDex. S.A.J.T. and C.B. have an intellectual property interest in detecting NAAD as a biomarker of elevated NAD+ metabolism. C.B. is the chief scientific adviser of ProHealthspan, which sells NR supplements. The remaining authors declare no competing financial interests.

