## [Peer review file · Nature Communications]

Reviewers' Comments:

Reviewer #1 (Remarks to the Author)

The authors have used targeted LC-MS to analyze the metabolism of nicotinamide riboside in humans and mice, identifying a novel route for nicotinic acid adenine dinucleotide. The analytical methodologies are well performed. Below are a few very minor comments:

- Ingested spelled incorrectly on supplemental tables.
- Supplemental fig1, again is y-axis mg wet weight, please specify in legend. Please explain why AUC for NR is negative in one case.
- The title ends in "mouse and man", I'm not sure if this is a reference to "of mice and men" but seems a little strange. Human would be better, especially since some of the individuals are women.
- Figure legends are numbered incorrectly.
- Figures 1-3, again is y-axis mg wet weight, please specify in legend.
- Figure 4 units?

Reviewer #2 (Remarks to the Author)

A. Summary of the key results

This study employed a mouse model and human objects to study whether and how may NR supplement impact NAD⁺ metabolome. In animal studies, additional NAD⁺ precursors such as NAM and NA were included for comparisons. By tracking the amount and time of appearance of key NAD⁺ metabolites in mouse liver (following NAD⁺ precursor treatments), preliminary pharmacokinetics of NR, NA and NAM were developed. Mouse pharmacokinetic data suggested that NR is a more efficient NAD⁺ precursor in liver. In human studies, NR showed a dose-dependent increase in blood cell NAD⁺ metabolome. This study also suggests NAAD is a sensitive biomarker for effective NR supplementation induced NAD⁺ production. Overall, data presented here support the claim that NR is bioavailable in human. However, to further support that NR is a more effective NAD⁺ precursor, additional studies in more cell/tissue types (at least in mouse) are required, given that different cells/tissue likely have different baseline activities of each NAD⁺ synthesis pathway. In some tissues, NR might be a more efficient NAD⁺ precursor whereas in others, NA or NAM might be more efficient.

B. Originality and interest: if not novel, please give references:

The novelty of this study lies in the use of human subjects to study the bioavailability and metabolism of orally administered NR as well as the development and comparisons of pharmacokinetics of different NAD⁺ precursors (NR, NA and NAM) in a mouse model.

C. Data & methodology: Most approaches appeared valid. Quality of presentation needs improvement. Figure numbers were wrong and some figure labels were not clear.

D. Appropriate use of statistics and treatment of uncertainties: Statistical analyses were adequate.

E. Conclusions: robustness, validity, reliability: additional studies are required to strengthen the major conclusions of this study.

F. Suggested improvements: see reviewer's suggestions listed below.

G. References: appropriate credit to previous work? Mostly.

H. Clarity and context: lucidity of abstract/summary. Appropriateness of abstract, introduction and conclusions: Abstract and Conclusions need to be modified to include additional possible interpretations of the results.

Reviewer's suggestions:

1. Figure 5-8 should have been Fig 1-4.
2. Some Y axis labels in Fig 4 did not show up properly in PDF format.
3. In Fig 2, the peaks of both NaMN and NAAD appeared at a time point before NAD⁺ increase following by NR treatment, suggesting a possible route of NR → Nam → NA → NAAD → NAD⁺. It is not clear how much NR is directly converted to NAD⁺ via NMN and how much NR is converted to NAD⁺ via other route(s).
4. It appeared that excess NAM is converted to MeNAM and Me4PY, which may reduce NAM induced cytotoxicity and help explain why NAM supplement does not lead to NAM accumulation in cells.
5. A significant fraction of supplemented NR is converted to NAM in mouse liver (Fig. 2d). The NAM peak appeared about the same time as the MeNam and Me4PY peaks (Fig 2d,e,f) suggesting excess NAM is converted to MeNam and Me4PY in mouse liver cells. In human blood cells (Fig 4d and 4e), it appeared that at higher dose of NR, the majority of NR was converted to NAM in the form of MeNAM and Me4PY. It is also possible that a fraction of NR is converted to Nam then NMN and finally NAD⁺. One major concern is that without discussions or results of kinetic parameters of key NAD⁺ biosynthetic enzymes, it is difficult to develop and compare pharmacokinetics for different NAD⁺ precursors.
6. Data presented In Figure 3 were not easy to comprehend in terms of how the % of single labeled or double-labeled NAD⁺ metabolome were calculated and what these results implied. The double-labeled NR (deuterium labeled ribose + ¹³C labeled Nam) experiments indeed showed that NAD⁺ metabolism is highly dynamic. Although the authors claimed that the % of double-labeled NAAD is similar to that of double labeled NAD⁺ and therefore, NAAD is likely resulted from NAD⁺. Without additional studies in different cell type(s) and taking into consideration of enzyme kinetics, it is difficult to draw a strong conclusion.

Reviewer #3 (Remarks to the Author)

The manuscript by Brenner and colleagues describes a tour de force study of the metabolism of the NAD precursor and vitamin nicotinamide riboside (NR). Nicotinic acid (Na) and nicotinamide (NaM) have been used for decades as vitamins and in the case of NA, as a cholesterol-lowering vitamin with flushing as a side effect that limits its use. The pharmacology and safety of NAD precursors is also of high interest given the increasing number of papers showing that NAD⁺ is a signaling molecule that activates cell defenses against aging and age-related diseases via PARPs and Sirtuins. The study initially looks at an n=1 in humans (which this reviewer at first thought was unsatisfactory) but then they perform an extensive and unique NAD metabolomics analysis of mice given NR. They use labeled NR to trace the metabolism of NR in mice and make a number of novel and very interesting discoveries. And finally test the effects of NR on a group of 12 volunteers to bolster the N=1 and the mouse work.

They show that in a limited trial size, there is no serious adverse effect in humans and that NR raises NAD in human blood samples. They also identify NAAD (nicotinic acid adenine dinucleotide) as a biomarker of Na (nic acid) synthesis from NR and surprisingly, Na, the only precursor expected to proceed to NAD through an NAAD intermediate, produced the least NAAD. They suggest that when NAD⁺ is elevated at least 2-fold, an unknown activity deamidates NAD⁺ to

NAAD. One intriguing possibility is that NAAD is formed by the long-sought enzyme that forms intracellular NAADP.

Major issues:

1. The data in figure 4 is extremely noisy, hence the desire of the authors to convert to AUC. Can the authors speculate on the causes of the variability? Could it be stress or a methodological variable? Perhaps another cell type/ tissue e.g. muscle would give less variable results. (Speaking of which, if the authors have muscle, fat and brain tissues those would be well worth processing.)
2. It is a shame the study didn't report changes in cholesterol or fasting blood glucose during the study. Liver enzymes or potassium. A washout period would be interesting to follow also. How long lasting are the effects.
3. Can the authors prove that NR is incorporated into NAAD after formation of NAD⁺ and chased back to the NAD⁺ peak as NAD⁺ declines? This is a major point in the paper.
4. Was there any degradation of the NR prior to injection. e.g. breakdown on NR to NaM? NR has some instability at room temp.
5. p.4. The authors write: "Based on the ability of NR to elevate NAD⁺ synthesis, increase sirtuin activity and extend lifespan in yeast^{6,22}". Here it is appropriate to cite Anderson et al., Nature 2003, who showed increased copies of NAD salvage pathway genes extends yeast lifespan and underlies the effect of CR. This was also shown to be true for Drosophila Balan et al, JBC 2007.
6. The authors write: "Because of the abundance of NAD-dependent processes, it is not known to what degree NAD boosting strategies are mechanistically dependent on particular molecules such as SIRT1 or SIRT3." Not true. Gomes et al Cell 2013 showed dependence of NMN on SIRT1.

Minor issues:

7. Consider plotting Table 1 as graphs in the main figures.
8. Provide the reference where "NR prevents pellagra"
9. P4. though its use is limited by painful flushing^{19,20}." Its not so much pain one feels, as it is discomfort.
10. It would be fascinating to test the effects in an older mouse that has lower NAD levels.
11. Figure legends need to be reduced by 4. Presumably the two manuscripts were once one.
12. P13. "In an accompanying project, we showed that hepatic cells convert NMN extracellularly to NR and that both NMN and NR depend on expression of NRK1 for conversion to cellular NAD." Should say "partially depend".
13. P5. human beings, consider writing "humans"
14. Was there any sex difference? If not, note that.

Reviewer #4 (Remarks to the Author)

The paper by Trammel S et al. reports on the metabolic fate of pharmacological doses of NA, NR and Nam orally administered to mouse and human. By profiling the NAD metabolome in mouse

liver and human PBMCs, Authors show that NA, NR and Nam elevate intracellular NAD with distinctive kinetics and differently affect the formation of ADPR, Nam and Nam catabolites, and that all three precursors lead to NAAD production. While previous reports have already shown that in the mammalian liver Nam and NA have distinct metabolic fates (Collins PB and Chaykin S, JBC 247, 778, 1972), this is the first report comparing the pharmacokinetics of the three vitamins and showing that NR has a distinctive bioavailability. Authors propose NAAD as a direct product of NR utilization and as a biomarker of boosting NAD.

The work raises several concerns:

Authors show that following administration of pharmacological amounts of the three forms of vitamin B3, a raise of both NAD and NAAD levels was observed in liver and PMCBs. Therefore they propose NAAD as a biomarker of boosting NAD. This referee believes that, in order to define NAAD as a biomarker, a wider range of organs and tissues should have been examined.

The interpretation of the data reported in the various panels of Figure 2 is in some cases conflicting. As an example: Authors show that differently from NA administration that does not affect hepatic Nam levels while increasing ADPR, Nam administration leads to the increase of Nam, but is ineffective on ADPR. Authors state that this is consistent with NA being better than Nam in promoting NAD catabolism since, differently from Nam, NA does not inhibit sirtuin activity. At the same time, Authors show that NR administration stimulates accumulation of both Nam and ADPR (figure 2d, 2j) and they ascribe this to "greater NAD consuming activities". However, the significant increase in Nam measured after NR gavage is likely to inhibit, rather than stimulate, NAD consumers and it is surely in contrast with the knowledge that sirtuins activity is required for NR-mediated liver benefit (Gariani K et al, Hepatology, 2015).

Some data presented in Figure 2 and 3 have not been discussed at all: this makes Authors' conclusions unconvincing. As examples: Authors show that NR gavage induces raises in NAD, NMN, NAAD and NAMN levels. They state that "production of high levels of NAAD from NAD could account for the NR-stimulated peak in NAMN because NAMN adenylyltransferase is known to be a reversible enzyme" (line 11, page 20). Since the enzyme NMNAT that catalyzes NAMN \leftrightarrow NAAD conversion is the same catalyzing NMN \leftrightarrow NAD conversion with comparable efficiency, and both reactions have similar equilibrium constant as shown by the finding that the NAD/NMN ratio is similar to NAAD/NAMN ratio in mice liver (Mori V et al, PlosOne 9, 2014), one would expect that the two ratios would remain similar after NR gavage. However by inspecting the data presented in Figure 2, ratios of about 400 and 40 can be calculated for NAD/NMN and NAAD/NAMN, respectively. This discrepancy strongly weakens the hypothesis that NAMN might derive from NAAD.

In addition, Authors' interpretation of data presented in Fig 3 appears rather biased. To validate their hypothesis on the occurrence of a retrograde pathway from NAD to NAAD, Authors highlighted the result that at 2 h NAAD and NAD share the same heavy atoms composition. However, such a similar heavy atoms composition is absent in the subsequent time points. Again, this result does not appear to support the Authors' hypothesis.

As for the experiment with double-labeled NR, which, as stated by the Authors, was performed with the same experimental design used in pharmacokinetic analyses, it is clear that the effect of the double-labeled NR administration on NAD levels is different from that of the unlabeled NR (Fig 3a versus Fig 2b). In the double-label experiment NAD doesn't peak: NAD level is already high at 2h and keeps increasing with time. The reason of this discrepancy between the two experiments should have been discussed, also in view of the finding that at 8h NAD continues to increase while NAAD has already dropped (Fig 3a and Fig 3e). This speaks against the hypothesis that increase of NAD from NR drives NAAD formation.

The finding that high dose Nam administration raises NAAD levels in liver is not novel. Collins PB

and Chaykin S (JBC 247, 778, 1972), by IP injection of high amounts of Nam in mice reported the formation of NAAD in the liver. Since Trammel et al. show that NR gavage leads to significant accumulation of hepatic Nam (Figure 2d), the occurrence of a raise in NAAD might have been anticipated. In their work, Collins PB and Chaykin S did not rule out the possibility that an enzyme able to deamidate Nam might initiate the deamidated pathway to NAAD. They referred to an enzyme purified from liver able to catalyze Nam deamidation with a Km in the range of the boosted Nam level. Authors shouldn't have ruled out this hypothesis.

As for the clinical study, here are minor points:

In Figure 4, labels of y-axes are not clear.

In the graph of Figure 4b, it is not shown the statistical significance of the raise in NAD.

Reviewer #5 (Remarks to the Author)

This study reveals very good bioavailability and metabolic efficacy of nicotinamide riboside (NR) given orally for modifying the NAD⁺ metabolome. A novel observation is the marked increase in nicotinic acid adenine dinucleotide (NAAD), which might serve as a biomarker for altered NAD⁺ metabolism.

The authors note that NAD⁺ and Nam levels in humans have relatively high variability. Is there data on intraindividual (day-to-day or week-to-week) versus interindividual variability? Regardless of how the variability arises, there may be a stronger rationale for using NAAD as a biomarker because of the variability of the known functional metabolites. This might deserve mention.

This paper resembles the reporting of a Phase 1 pharmacodynamic study for a new drug. As such, the paper provides needed data on NR metabolism. However, since NR is a dietary supplement by FDA rules, we may be missing some critical data as NR may be increasingly targeted as a pharmacologic type of intervention, even by lay people without medical supervision. Are there any small animal toxicity studies for NR? Is there any data on potential human toxicity, such as AST assays? If not, it is imperative to point out the lack of such safety data.

At present, results suggesting beneficial pharmacologic effects for NR are thin, although the authors in earlier work have outlined some potential benefits.

I am surprised that the authors cite no results on plasma lipoprotein changes from administration of NR to mice. Very early niacin research showed the absence of cholesterol lowering by nicotinamide, compared to effective cholesterol lowering by nicotinic acid. It is true that Merck scientists recently found no or little role for GPR109A (now called HCA2 for hydroxycarboxylic acid receptor-2) in lipoprotein changes by nicotinic acid, but the comparative effectiveness of nicotinic acid versus nicotinamide (versus NR?) for modifying lipoprotein levels seems not to have explored with modern techniques.

In any case, the Discussion in this paper should acknowledge the lack of efficacy of nicotinamide for modifying lipoprotein levels. I believe that Altschul initially made this observation, and I believe L A Carlson cited it in his 50th anniversary review on niacin.

Serious liver toxicity, including one case of requirement for liver transplant, has followed inappropriate use of slow-release nicotinic acid. See Guyton and Bays, Am J Cardiol 2007. The fact that NAD⁺ metabolism may be altered more by Nam and by NR than by NA raises concern for potential liver toxicity by Nam and NR.

In this regard, the 7-day dosing of NR 1 g daily reverses the usual order for safe examination of a new drug. The fact that legally NR is considered a dietary supplement does not argue against

keeping the order of studies initially with single doses and subsequently multiple day dosing. I believe the authors should leave out all reference to the 7 day dose study, because it is not essential to the findings and because it gives the impression especially to lay persons and news reporters that the compound is known to be safe.

When multiple day dose studies are performed, they should be done with appropriate safety screening, including prior toxicity studies in small animals.

Summary. "Blood cell," please consider "peripheral mononuclear blood cell."

The use of molar concentrations for metabolites in PBMC preparations should be explained in detail in Methods. PBMCs account for only a tiny fraction of total blood volume. Therefore, the denominator for these assays could be subject to error.

Page 7. "NR can be phosphorylated to Nam by purine nucleoside phosphorylase and still contribute to NAD⁺ synthesis through Nam salvage^{22,28}." Please explain how nicotinamide riboside, which contains no phosphate group, can be "phosphorylated" (dephosphorylated?). Why is the alternative pathway in yeast relevant at this point?

Page 8. "...the rise in PBMC NAD⁺ was not as pronounced as the spike in NAAD" should read "...the relative rise..."

Page 8 "...are found exclusively in PBMCs" Please re-phrase, since red blood cells, platelets, and polymorphonuclear leukocytes were not examined.

Page 12 "capable of improving reverse cholesterol transport" Reverse cholesterol transport is an extraordinarily complicated physiologic phenomenon. Effects in the periphery rather than in the liver may be regulatory for reverse cholesterol transport. Do the authors mean here "capable of increasing high density lipoprotein cholesterol" or "capable of increasing high density lipoprotein production" or what?

Page 15. "At 500 mg of niacin" Is this 300?

Page 17. It would be helpful to know the percent increases in AUC above background AUC (i.e., baseline value x 24 h).

Figure 4. It appears that the levels of NAD⁺ and Nam are fairly well defended in PBMCs, since NAD⁺ increase was marginally significant, and Nam increase was not significant, but the levels of metabolic conversion products did increase.

Page 17. "NMN (Fig. 4a) and NAD⁺ (Fig. 4b) rose to higher concentrations of ~2 μ M and 20 μ M, respectively, in people given 300 mg and 1000 mg doses of NR than in people given 100 mg doses of NR." This is not correct for NMN, since the overall increase was not statistically significant. The increase in NAD⁺ should be characterized as percent increase in lieu of or in addition to the absolute increase.

Reference 24. In press 2014?

Legends. Please turn off consecutive numbering for Figure X.

June 9, 2016

Dear Dr. Schmitt,

Here are our point-by-point responses to review.

Reviewer #1 (Expert in metabolomics; Remarks to the Author):

The authors have used targeted LC-MS to analyze the metabolism of nicotinamide riboside in humans and mice, identifying a novel route for nicotinic acid adenine dinucleotide. The analytical methodologies are well performed. Below are a few very minor comments:

We thank the reviewer for appreciating the novelty.

-Ingested spelled incorrectly on supplemental tables.

Corrected.

-Supplemental fig1, again is y-axis mg wet weight, please specify in legend. Please explain why AUC for NR is negative in one case.

Specified. In a baseline-subtracted AUC, values can be negative. NR is a low abundance metabolite and its levels do not respond to supplementation. In this experiment, the zero time observation was around 50 pmol/mg wet weight. Four subsequent measurements were below this value and two were above. The baseline-subtracted AUC is effectively zero.

-The title ends in "mouse and man", I'm not sure if this is a reference to "of mice and men" but seems a little strange. Human would be better, especially since some of the individuals are women.

Corrected.

-Figure legends are numbered incorrectly.

Corrected.

-Figures 1-3, again is y-axis mg wet weight, please specify in legend.

Specified.

-Figure 4 units?

Corrected.

Reviewer #2 (Expert in NAD metabolism; Remarks to the Author):

A. Summary of the key results

This study employed a mouse model and human objects to study whether and how may NR supplement impact NAD⁺ metabolome. In animal studies, additional NAD⁺ precursors such as NAM and NA were included for comparisons. By tracking the amount and time of appearance of key NAD⁺ metabolites in mouse liver (following NAD⁺ precursor treatments), preliminary pharmacokinetics of NR, NA and NAM were developed. Mouse pharmacokinetic data suggested that NR is a more efficient NAD⁺ precursor in liver. In human studies, NR showed a dose-dependent increase in blood cell NAD⁺ metabolome. This study also suggests NAAD is a sensitive biomarker for effective NR supplementation induced NAD⁺ production. Overall, data presented here support the claim that NR is bioavailable in human. However, to further support that NR is a more effective NAD⁺ precursor, additional studies in more cell/tissue types (at least in mouse) are required, given that different cells/tissue likely have different baseline activities of each NAD⁺ synthesis pathway. In some tissues, NR might be a more efficient NAD⁺ precursor whereas in others, NA or NAM might be more efficient.

We agree. We provide data from heart in this revision.

B. Originality and interest: if not novel, please give references:

The novelty of this study lies in the use of human subjects to study the bioavailability and

metabolism of orally administered NR as well as the development and comparisons of pharmacokinetics of different NAD⁺ precursors (NR, NA and NAM) in a mouse model.

We agree.

C. Data & methodology: Most approaches appeared valid. Quality of presentation needs improvement. Figure numbers were wrong and some figure labels were not clear.

We corrected figure numbers and clarified labels.

D. Appropriate use of statistics and treatment of uncertainties: Statistical analyses were adequate.

OK.

E. Conclusions: robustness, validity, reliability: additional studies are required to strengthen the major conclusions of this study.

Two additional figures and a table are included to strengthen conclusions.

F. Suggested improvements: see reviewer's suggestions listed below.

Noted below.

G. References: appropriate credit to previous work? Mostly.

Several referenced added.

H. Clarity and context: lucidity of abstract/summary. Appropriateness of abstract, introduction and conclusions: Abstract and Conclusions need to be modified to include additional possible interpretations of the results.

We made these changes.

Reviewer's suggestions:

1. Figure 5-8 should have been Fig 1-4.

Corrected.

2. Some Y axis labels in Fig 4 did not show up properly in PDF format.

Corrected.

3. In Fig 2, the peaks of both NaMN and NAAD appeared at a time point before NAD⁺ increase following by NR treatment, suggesting a possible route of NR → Nam → NA → NAAD → NAD⁺. It is not clear how much NR is directly converted to NAD⁺ via NMN and how much NR is converted to NAD⁺ via other route(s).

The peaks of NAMN (15 pmol/mg) and NAAD (400 pmol/mg) are at 4 hours, whereas the peak of NAD⁺ (> 4000 pmol/mg) is at 6 hours. It is important to emphasize the absolute size of these peaks were measured by quantitative targeted metabolomics. Gavage of NA produced a 400 pmol/mg peak of NA. However, neither gavage of NR nor Nam produced any peak of NA, which eliminates the possibility that a 400 pmol/mg peak of NAAD came from NA. Additional work in this paper shows that gavage of double labeled NR produces double labeled NAD⁺, NADP⁺ and NAAD, prior to the rise in total NAD⁺ but it also produces a great deal of single labeled NMN and NAD⁺. This is because pushing NR into the NAD⁺ pool drives production of Nam, ADPR and NAAD and ultimately leads to return to a baseline level of NAD⁺. With excellent direction from reviewers, we made these points much clearer in the revision.

4. It appeared that excess NAM is converted to MeNAM and Me4PY, which may reduce NAM induced cytotoxicity and help explain why NAM supplement does not lead to NAM accumulation in cells.

Yes.

5. A significant fraction of supplemented NR is converted to NAM in mouse liver (Fig. 2d). The NAM peak appeared about the same time as the MeNam and Me4PY peaks (Fig 2d,e,f) suggesting excess NAM is converted to MeNam and Me4PY in mouse liver cells. In human blood cells (Fig 4d and 4e), it appeared that at higher dose of NR, the majority of NR was converted to NAM in the form of MeNAM and Me4PY. It is also possible that a fraction of NR is converted to

Nam then NMN and finally NAD⁺. One major concern is that without discussions or results of kinetic parameters of key NAD⁺ biosynthetic enzymes, it is difficult to develop and compare pharmacokinetics for different NAD⁺ precursors.

This body of work represents the most comprehensive analysis of NR versus Nam versus NA in any experimental system.

6. Data presented in Figure 3 were not easy to comprehend in terms of how the % of single labeled or double-labeled NAD⁺ metabolome were calculated and what these results implied. The double-labeled NR (deuterium labeled ribose + ¹³C labeled Nam) experiments indeed showed that NAD⁺ metabolism is highly dynamic. Although the authors claimed that the % of double-labeled NAAD is similar to that of double labeled NAD⁺ and therefore, NAAD is likely resulted from NAD⁺. Without additional studies in different cell type(s) and taking into consideration of enzyme kinetics, it is difficult to draw a strong conclusion.

The experiments definitively show that NR is incorporated intact into NAD⁺, NADP⁺ and NAAD in the liver and that NAAD is a sensitive biomarker of elevating NAD metabolism in mouse heart and liver and in human blood.

Reviewer #3 (Expert in NAD metabolism; Remarks to the Author):

The manuscript by Brenner and colleagues describes a tour de force study of the metabolism of the NAD precursor and vitamin nicotinamide riboside (NR). Nicotinic acid (Na) and nicotinamide (NaM) have been used for decades as vitamins and in the case of NA, as a cholesterol-lowering vitamin with flushing as a side effect that limits its use. The pharmacology and safety of NAD precursors is also of high interest given the increasing number of papers showing that NAD⁺ is a signaling molecule that activates cell defenses against aging and age-related diseases via PARPs and Sirtuins. The study initially looks at an n=1 in humans (which this reviewer at first thought was unsatisfactory) but then they perform an extensive and unique NAD metabolomics analysis of mice given NR. They use labeled NR to trace the metabolism of NR in mice and make a number of novel and very interesting discoveries. And finally test the effects of NR on a group of 12 volunteers to bolster the N=1 and the mouse work.

They show that in a limited trial size, there is no serious adverse effect in humans and that NR raises NAD in human blood samples. They also identify NAAD (nicotinic acid adenine dinucleotide) as a biomarker of Na (nic acid) synthesis from NR and surprisingly, Na, the only precursor expected to proceed to NAD through an NAAD intermediate, produced the least NAAD. They suggest that when NAD⁺ is elevated at least 2-fold, an unknown activity deamidates NAD⁺ to NAAD. One intriguing possibility is that NAAD is formed by the long-sought enzyme that forms intracellular NAADP.

Major issues:

1. The data in figure 4 is extremely noisy, hence the desire of the authors to convert to AUC. Can the authors speculate on the causes of the variability? Could it be stress or a methodological variable? Perhaps another cell type/ tissue e.g. muscle would give less variable results. (Speaking of which, if the authors have muscle, fat and brain tissues those would be well worth processing.)

The human n=12 study only collected blood and urine. People are inherently variable.

2. It is a shame the study didn't report changes in cholesterol or fasting blood glucose during the study. Liver enzymes or potassium. A washout period would be interesting to follow also. How long lasting are the effects.

Single doses of NR would not be expected to improve parameters of metabolic disease.

Moreover, the volunteers were lean and not on medication for cholesterol. Washouts were done between doses. The time-dependent effects of NR on human blood NAD⁺ metabolism are shown in Figure 2.

3. Can the authors prove that NR is incorporated into NAAD after formation of NAD⁺ and chased back to the NAD⁺ peak as NAD⁺ declines? This is a major point in the paper.

We proved that NR is incorporated into NAD⁺ and NAAD in the same proportions and that incorporation into both NAD⁺ and NAAD occurs prior to the steady-state rise in NAD⁺.

4. Was there any degradation of the NR prior to injection. e.g. breakdown on NR to NaM? NR has some instability at room temp.

No. All of our materials are mass-spec validated and, in fact, the precise methods in this paper should improve best practices in handling NR and performing quantitative NAD metabolomics.

5. p.4. The authors write: "Based on the ability of NR to elevate NAD⁺ synthesis, increase sirtuin activity and extend lifespan in yeast^{6,22}". Here it is appropriate to cite Anderson et al., Nature 2003, who showed increased copies of NAD salvage pathway genes extends yeast lifespan and underlies the effect of CR. This was also shown to be true for Drosophila Balan et al, JBC 2007.

6. The authors write: "Because of the abundance of NAD-dependent processes, it is not known to what degree NAD boosting strategies are mechanistically dependent on particular molecules such as SIRT1 or SIRT3." Not true. Gomes et al Cell 2013 showed dependence of NMN on SIRT1.

We added several references to activities of NR and evidence for NR and NMN as sirtuin activating compounds.

Minor issues:

7. Consider plotting Table 1 as graphs in the main figures.

Excellent suggestion. This is now Figure 2.

8. Provide the reference where "NR prevents pellagra"

We didn't say that NR prevents pellagra.

9. P4. though its use is limited by painful flushing^{19,20}." Its not so much pain one feels, as it is discomfort.

Many people who have discontinued high dose niacin find it painful and this terminology is found in the literature.

10. It would be fascinating to test the effects in an older mouse that has lower NAD levels.

We agree but this is beyond the scope of this study.

11. Figure legends need to be reduced by 4. Presumably the two manuscripts were once one.

The figure legends have been corrected. There was auto-numbering between figure legends and numbers that caused the error.

12. P13. "In an accompanying project, we showed that hepatic cells convert NMN extracellularly to NR and that both NMN and NR depend on expression of NRK1 for conversion to cellular NAD." Should say "partially depend".

The two papers are stand-alone and there is no longer reference to any unpublished work.

13. P5. human beings, consider writing "humans"

Changed to people.

14. Was there any sex difference? If not, note that.

The revision notes that there were no sex differences.

Reviewer #4 (Expert in NAD metabolism; Remarks to the Author):

The paper by Trammel S et al. reports on the metabolic fate of pharmacological doses of NA, NR

and Nam orally administered to mouse and human. By profiling the NAD metabolome in mouse liver and human PBMCs, Authors show that NA, NR and Nam elevate intracellular NAD with distinctive kinetics and differently affect the formation of ADPR, Nam and Nam catabolites, and that all three precursors lead to NAAD production. While previous reports have already shown that in the mammalian liver Nam and NA have distinct metabolic fates (Collins PB and Chaykin S, JBC 247, 778, 1972), this is the first report comparing the pharmacokinetics of the three vitamins and showing that NR has a distinctive bioavailability. Authors propose NAAD as a direct product of NR utilization and as a biomarker of boosting NAD.

The work raises several concerns:

Authors show that following administration of pharmacological amounts of the three forms of vitamin B3, a raise of both NAD and NAAD levels was observed in liver and PMCBs. Therefore they propose NAAD as a biomarker of boosting NAD. This referee believes that, in order to define NAAD as a biomarker, a wider range of organs and tissues should have been examined.

The paper now analyzes liver and heart NAD+ metabolomes in mouse plus blood and urine NAD+ metabolomes in people.

The interpretation of the data reported in the various panels of Figure 2 is in some cases conflicting. As an example: Authors show that differently from NA administration that does not affect hepatic Nam levels while increasing ADPR, Nam administration leads to the increase of Nam, but is ineffective on ADPR. Authors state that this is consistent with NA being better than Nam in promoting NAD catabolism since, differently from Nam, NA does not inhibit sirtuin activity. At the same time, Authors show that NR administration stimulates accumulation of both Nam and ADPR (figure 2d, 2j) and they ascribe this to "greater NAD consuming activities". However, the significant increase in Nam measured after NR gavage is likely to inhibit, rather than stimulate, NAD consumers and it is surely in contrast with the knowledge that sirtuins activity is required for NR-mediated liver benefit (Gariani K et al, Hepatology, 2015).

The data show that Nam is further processed to me-Nam, me2PY and me4PY.

Some data presented in Figure 2 and 3 have not been discussed at all: this makes Authors' conclusions unconvincing. As examples: Authors show that NR gavage induces raises in NAD, NMN, NAAD and NAMN levels. They state that "production of high levels of NAAD from NAD could account for the NR-stimulated peak in NAMN because NAMN adenyltransferase is known to be a reversible enzyme" (line 11, page 20). Since the enzyme NMNAT that catalyzes NAMN \leftrightarrow NAAD conversion is the same catalyzing NMN \leftrightarrow NAD conversion with comparable efficiency, and both reactions have similar equilibrium constant as shown by the finding that the NAD/NMN ratio is similar to NAAD/NAMN ratio in mice liver (Mori V et al, PlosOne 9, 2014), one would expect that the two ratios would remain similar after NR gavage. However by inspecting the data presented in Figure 2, ratios of about 400 and 40 can be calculated for NAD/NMN and NAAD/NAMN, respectively. This discrepancy strongly weakens the hypothesis that NAMN might derive from NAAD.

In addition, Authors' interpretation of data presented in Fig 3 appears rather biased. To validate their hypothesis on the occurrence of a retrograde pathway from NAD to NAAD, Authors highlighted the result that at 2 h NAAD and NAD share the same heavy atoms composition. However, such a similar heavy atoms composition is absent in the subsequent time points. Again, this result does not appear to support the Authors' hypothesis.

As for the experiment with double-labeled NR, which, as stated by the Authors, was performed with the same experimental design used in pharmacokinetic analyses, it is clear that the effect of the double-labeled NR administration on NAD levels is different from that of the unlabeled NR (Fig 3a

versus Fig 2b). In the double-label experiment NAD doesn't peak: NAD level is already high at 2h and keeps increasing with time. The reason of this discrepancy between the two experiments should have been discussed, also in view of the finding that at 8h NAD continues to increase while NAAD has already dropped (Fig 3a and Fig 3e). This speaks against the hypothesis that increase of NAD from NR drives NAAD formation.

The reviewer raises several good points. We carefully revised the manuscript to clarify and distinguish facts and interpretations. The paper clearly shows that hepatic NAD⁺, NADP and NAAD incorporate both the ribose and the nicotinamide from NR. We further showed that neither NR nor Nam is converted to NAD⁺ through a NA intermediate. We showed that the rise of NAAD is largely coincident with the rise in NAD⁺ and that liver incorporates double-labeled NR into double labeled NAD⁺ and NAAD prior to a rise in the total levels of NAD⁺.

The finding that high dose Nam administration raises NAAD levels in liver is not novel. Collins PB and Chaykin S (JBC 247, 778, 1972), by IP injection of high amounts of Nam in mice reported the formation of NAAD in the liver. Since Trammel et al. show that NR gavage leads to significant accumulation of hepatic Nam (Figure 2d), the occurrence of a raise in NAAD might have been anticipated. In their work, Collins PB and Chaykin S did not rule out the possibility that an enzyme able to deamidate Nam might initiate the deamidated pathway to NAAD. They referred to an enzyme purified from liver able to catalyze Nam deamidation with a Km in the range of the boosted Nam level. Authors shouldn't have ruled out this hypothesis.

The reviewer pointed out important work for us to cite. We have included this citation and several additional citations. However, our work is strikingly novel. This body of work shows that Nam and NR do not form NA on the way to forming NAD⁺, so that is not the mechanism. Double-labeled NR was used to show that both moieties of NR are incorporated into NAD⁺ even before the total concentration of NAD⁺ goes up, that NAAD incorporates the double label, and that formation of newly labeled NAD⁺ and NAAD is accompanied by production of single labeled species. All of this was done with oral NR at the same effective dose as taken by people and much of the experiments were done side-by-side with the two other NAD⁺ precursor vitamins.

As for the clinical study, here are minor points:

In Figure 4, labels of y-axes are not clear.

In the graph of Figure 4b, it is not shown the statistical significance of the raise in NAD.

We thank the reviewer for these points. We have made the necessary corrections.

Reviewer #5 (Clinician; Remarks to the Author):

This study reveals very good bioavailability and metabolic efficacy of nicotinamide riboside (NR) given orally for modifying the NAD⁺ metabolome. A novel observation is the marked increase in nicotinic acid adenine dinucleotide (NAAD), which might serve as a biomarker for altered NAD⁺ metabolism.

We thank the review for this summary.

The authors note that NAD⁺ and Nam levels in humans have relatively high variability. Is there data on intraindividual (day-to-day or week-to-week) versus interindividual variability? Regardless of how the variability arises, there may be a stronger rationale for using NAAD as a biomarker because of the variability of the known functional metabolites. This might deserve mention.

Good points for a follow-up study.

This paper resembles the reporting of a Phase 1 pharmacodynamic study for a new drug. As such, the paper provides needed data on NR metabolism. However, since NR is a dietary supplement by FDA rules, we may be missing some critical data as NR may be increasingly targeted as a

pharmacologic type of intervention, even by lay people without medical supervision. Are there any small animal toxicity studies for NR? Is there any data on potential human toxicity, such as AST assays? If not, it is imperative to point out the lack of such safety data.

The revised manuscript includes a citation of a rat toxicology study. NR was found to be as safe as Nam.

At present, results suggesting beneficial pharmacologic effects for NR are thin, although the authors in earlier work have outlined some potential benefits.

I am surprised that the authors cite no results on plasma lipoprotein changes from administration of NR to mice. Very early niacin research showed the absence of cholesterol lowering by nicotinamide, compared to effective cholesterol lowering by nicotinic acid. It is true that Merck scientists recently found no or little role for GPR109A (now called HCA2 for hydroxycarboxylic acid receptor-2) in lipoprotein changes by nicotinic acid, but the comparative effectiveness of nicotinic acid versus nicotinamide (versus NR?) for modifying lipoprotein levels seems not to have explored with modern techniques.

The revision cites work showing that NR lowers cholesterol in overfed mice.

In any case, the Discussion in this paper should acknowledge the lack of efficacy of nicotinamide for modifying lipoprotein levels. I believe that Altschul initially made this observation, and I believe L A Carlson cited it in his 50th anniversary review on niacin.

Cited.

Serious liver toxicity, including one case of requirement for liver transplant, has followed inappropriate use of slow-release nicotinic acid. See Guyton and Bays, Am J Cardiol 2007. The fact that NAD⁺ metabolism may be altered more by Nam and by NR than by NA raises concern for potential liver toxicity by Nam and NR.

There's no evidence that increased NAD⁺ metabolism is toxic. One of the net products of NAD⁺ metabolism, meNam, is a SIRT1 protein stabilizer in liver.

In this regard, the 7-day dosing of NR 1 g daily reverses the usual order for safe examination of a new drug. The fact that legally NR is considered a dietary supplement does not argue against keeping the order of studies initially with single doses and subsequently multiple day dosing. I believe the authors should leave out all reference to the 7 day dose study, because it is not essential to the findings and because it gives the impression especially to lay persons and news reporters that the compound is known to be safe.

When multiple day dose studies are performed, they should be done with appropriate safety screening, including prior toxicity studies in small animals.

The revised manuscript cites a rat toxicology study showing that NR is safe.

Summary. "Blood cell," please consider "peripheral mononuclear blood cell."

Done.

The use of molar concentrations for metabolites in PBMC preparations should be explained in detail in Methods. PBMCs account for only a tiny fraction of total blood volume. Therefore, the denominator for these assays could be subject to error.

The field is moving toward analyzing NAD⁺ metabolomes from whole blood such that molar with respect to blood volume is definitely the way to go. That's the way we reported concentrations in this study and, as the reviewer points out, this avoids dividing by a small volume, which would be subject to large errors.

Page 7. "NR can be phosphorylated to Nam by purine nucleoside phosphorylase and still contribute to NAD⁺ synthesis through Nam salvage^{22,28}." Please explain how nicotinamide riboside, which contains no phosphate group, can be "phosphorylated" (dephosphorylated?). Why is the alternative pathway in yeast relevant at this point?

Phosphorylyzed means that a covalent bond is broken by addition of a phosphate. The enzyme that catalyzes this (purine nucleoside phosphorylase) is found in yeast and mammals. Page 8. "...the rise in PBMC NAD⁺ was not as pronounced as the spike in NAAD" should read "...the relative rise..."

Corrected.

Page 8 "...are found exclusively in PBMCs" Please re-phrase, since red blood cells, platelets, and polymorphonuclear leukocytes were not examined.

Corrected.

Page 12 "capable of improving reverse cholesterol transport" Reverse cholesterol transport is an extraordinarily complicated physiologic phenomenon. Effects in the periphery rather than in the liver may be regulatory for reverse cholesterol transport. Do the authors mean here "capable of increasing high density lipoprotein cholesterol" or "capable of increasing high density lipoprotein production" or what?

Corrected.

Page 15. "At 500 mg of niacin" Is this 300?

Correct as written.

Page 17. It would be helpful to know the percent increases in AUC above background AUC (i.e., baseline value x 24 h).

We created a new Supplementary Table to illustrate this point. As the reviewer might have guessed, the percentage increase in NAAD AUC is much greater than the percentage increase in NAD⁺ AUC.

Figure 4. It appears that the levels of NAD⁺ and Nam are fairly well defended in PBMCs, since NAD⁺ increase was marginally significant, and Nam increase was not significant, but the levels of metabolic conversion products did increase.

Exactly.

Page 17. "NMN (Fig. 4a) and NAD⁺ (Fig. 4b) rose to higher concentrations of ~2 μ M and 20 μ M, respectively, in people given 300 mg and 1000 mg doses of NR than in people given 100 mg doses of NR." This is not correct for NMN, since the overall increase was not statistically significant. The increase in NAD⁺ should be characterized as percent increase in lieu of or in addition to the absolute increase.

We corrected the text.

Reference 24. In press 2014?

We corrected the citation.

Legends. Please turn off consecutive numbering for Figure X.

We corrected the figure numbering.

We truly appreciate all of the time reviewers took to review and help us perfect this study.

Charles Brenner

Reviewers' Comments:

Reviewer #2 (Remarks to the Author)

The authors have addressed most of my questions and concerns.
One minor suggestion: Although the reviewer generally agreed with author's conclusions on Fig. 5, It was difficult for the reviewer to draw conclusions based on the explanations given in the figure legends. The reviewer had the same issues previously (old Fig.3). More explanations in the figure legends will be helpful.

Reviewer #3 (Remarks to the Author)

Reviewer #4 (Remarks to the Author)

This reviewer believes that the points previously raised have been properly addressed by the Authors. In the revised version additional data have been provided to support the conclusions and the results have been more thoroughly discussed.

Reviewer #5 (Remarks to the Author)

The authors are commended for their effective response to my comments and for their comprehensive evaluation of NAD⁺ metabolomics/kinetics in mice and humans. John R. Guyton MD